# Incorporating Location Aspects in Process Integration Methodology

**Hür Bütün \*, Ivan Kantor and François Maréchal**

Industrial Process and Energy Systems Engineering (IPESE), École Polytechnique Fédérale de Lausanne, 1951 Sion, Switzerland

\* Correspondence: hur.butun@epfl.ch

**Abstract:** The large potential for waste resource and heat recovery in industry has been motivating research toward increasing efficiency. Process integration methods have proven to be effective tools in improving industrial sites while decreasing their resource and energy consumption; however, location aspects and their impact are generally overlooked. This paper presents a method based on process integration, which considers the location of plants. The impact of the locations is included within the mixed integer linear programming framework in the form of heat losses, temperature and pressure drop, and piping cost. The objective function is selected as minimisation of the total cost of the system excluding piping cost and $\epsilon$-constraints are applied on the piping cost to systematically generate multiple solutions. The method is applied to a case study with industrial plants from different sectors. First, the interaction between two plants and their utility integration are illustrated, depending on the piping cost limit which results in the heat pump and boiler on one site being gradually replaced by excess heat recovered from the other plant. Then, the optimisation of the whole system is carried out, as a large-scale application. At low piping cost allowances, heat is shared through high pressure steam in above-ground pipes, while at higher piping cost limits the system switches toward lower pressure steam sharing in underground pipes. Compared to the business-as-usual operation of the sites, the optimal solution obtained with the proposed method leads to 20% reduction in the overall cost of the system, including the piping cost. Further reduction in the cost is possible using a state of the art method but the technical and economic feasibility is not guaranteed. Thus, the present work provides a tool to find optimal industrial symbiosis solutions under different investment limits on the infrastructure between plants.

**Keywords:** industrial excess heat; process integration; location aspects; piping; heat losses; mixed integer linear programming

---

## 1. Introduction

Initially motivated by fluctuating energy prices, and more recently by environmental concerns, energy efficiency remains a focus of regulations, such as the Europe 2020 goals [1]. Industrial energy consumption accounts for 40% of the world [2] and 26% of the European energy consumption [3], making it one of the key sectors for increasing energy efficiency. Energy consumption in industry is mostly in the form of heat; thus, energy efficiency improvements imposed by these regulations can only be achieved by using heat more efficiently within the system.

Waste heat in industry is often defined as heat discharged to environment from the cooling systems (e.g., cooling towers) as well as the energy conversion technologies (e.g., boilers) in the form of heat losses. However, according to Bendig et al. [4], this is classified as excess heat and waste heat is only the part which cannot be recovered within the process, by another process or by using an energy conversion system. This convention is used throughout this work.

According to Reference [5], industrial excess heat accounts for 5–30% of the industrial energy consumption in different countries, averaging 22% in the EU which corresponds to 5–6% of the overall consumption. There are several options for valorisation of excess heat including direct heat recovery within a process, integration of energy conversion technologies (e.g., organic Rankine cycles) and heat recovery through other processes. Bendig et al. [4] suggested that a hierarchy is required between those options. Direct heat recovery is the most preferable since it typically requires the least investment and yields the largest improvement. Following this, remaining heat can be upgraded by heat pumps (HPs), transferred to another process or converted to another form, for example by using an organic Rankine cycle.

The International Energy Agency (IEA) classifies excess heat potential into theoretical, technical and economic potential [6]. Theoretical potential corresponds to the thermodynamic potential without considering the technologies for heat recovery. Technical potential takes into account the availability of technologies for heat recovery. For example, although the steel industry has large heat losses at high temperatures, the technical potential is low since technologies to recover heat from solids are not well developed. Finally, economic potential, leading to the heat recovery options the industries would be willing to invest in, accounts for the cost of heat recovery. Thus, when energy efficiency improvement options are considered, it is crucial to assess the technical and economic feasibility.

This paper, motivated by the high excess heat potential in industry and the importance of identifying economically feasible solutions, presents a novel methodology to determine heat and resource recovery within and between industrial processes. Instead of imposing a predefined hierarchy between the heat recovery options, the method introduces location aspects in process integration (PI) to obtain the optimal path for heat recovery under different investment cost limits on piping. Section 2 covers the methods available in the literature for improving industrial energy efficiency, Section 3 explains the method by going through the formulation in detail, Section 4 presents the case study that is used as a proof of concept, Section 5 discusses the results and Section 6 draws the conclusions of this work.

## 2. State of the Art

PI is a domain in energy efficiency research which aims to increase the heat and material recovery between processes and therefore reduce external resource dependency and energy supplied by fossil fuel-based technologies such as natural gas boilers. PI has been a research-intensive field since the oil crisis in the early 1970s. Although industrial energy efficiency was the main motivation for PI, it has also been used in urban energy system design [7], biomass conversion systems [8] and large-scale energy planning. The methods used in PI are classified in two main groups—namely graphical and mathematical programming (MP) methods.

Graphical methods in PI are based on pinch analysis (PA), which divide the system into hot (i.e., heat source) and cold streams (i.e., heat sink), aiming to maximise the heat exchange between them to minimise the hot and cold utility requirements. PA was first developed by Linnhoff and Hindmarsh [9] for a single industrial process. Afterwards, it was extended to total site analysis (TSA) in which the system consists of several production processes [10]. In PA, direct heat exchange between processes is considered. However, such exchanges may be problematic because of startup and shutdown dynamics or plant layout. Using utility systems for exchanges between the processes can solve those problems, since they have more operational flexibility. Hui and Ahmad [11] proposed a TSA method using utility systems. They considered the use of steam as an intermediate fluid for the exchange between processes. The capital cost of the heat exchanger network (HEN) was also included in their analysis while being ignored in the previous TSA methods. The design of utility systems is crucial in TSA as they commonly include centralised supply of heat and power to several plants. Pirmohamadi et al. [12] studied the optimal design of cogeneration systems in total sites. The method was based on site utility grand composite curves and aimed at maximising the exergy efficiency of the overall system. Short-term and seasonal storage play an important role in inter-plant heat recovery

in the case of multi-period problems. Liew et al. [13] extended the TSA methodology into seasonal total site heat storage cascade to model energy flows between sites and storage systems to determine the required storage size. Exchange between multiple plants brings about other challenges, such as process control and safety. Song et al., proposed a strategy to divide large-scale TSA problems into smaller sections to cope with these issues [14]. TSA was applied in each section to obtain the total inter-plant heat recovery. In inter-plant heat exchange, connections between plants can be in different configurations such as series, parallel or split. In their TSA-based method, Wang et al. [15] identified the excess heat of the plants and analysed inter-plant recovery using different connection patterns. The parallel pattern yielded higher heat recovery, while coming at a higher investment cost.

When the excess heat from plants is identified manually, it is critical to decide which streams participate in inter-plant transfer. A strategy to select such streams was presented by Song et al. [16]. They also introduced the concept of inter-plant shifted composite curves to maximise heat recovery using minimum heat capacity flowrate intermediate circuits. Hackl et al. [17] studied heat recovery in industrial clusters using TSA and intermediate fluids. The energy consumption of the cluster was reduced by introducing a hot water loop between plants. While TSA helps to identify the targets of energy requirements of multiple processes/plants, it brings about challenges in implementation due to the variety of plants/companies involved in the exchange. A method to overcome such challenges was developed by Hackl and Harvey [18]. In the first step of their method, TSA was used to find the total site targets, while in the second step the number of plants/companies involved in inter-plant heat integration was minimised and the investment required for the integration was split into periods. Industrial excess heat can also be valorised in a district heating network (DHN) as well as other plants. Morandin et al. [19] considered a case with an industrial cluster and a DHN. They concluded that cluster-wide heat collection yields better integration with the DHNs than connecting each site individually. Although using TSA energy targets for several plants have been identified, most methods ignore the distance between them. Chew et al. [20] listed layout as one of the main issues in implementing total site heat integration. They also recommended including piping cost for better analysis of inter-plant heat integration [17] and performing heat recovery through DHNs [19]. Liew et al. [21] added layout aspects in TSA by considering heat losses, temperature and pressure drop. First, the heat cascade was constructed using the problem table method of [9]. Afterwards, the corresponding heat losses, pressure and temperature drop were calculated and the streams in the problem table method were corrected accordingly. Finally, the heat cascade re-formulated with the new temperatures and heat loads.

Even though PA-based methods are effective in obtaining targets for total sites, when the number of plants and utility systems increase, they generally fail to obtain optimal solutions [15]. MP-based methods emerged to fill this gap and now dominate the field. Most of the early work focused on utility integration [22] and heat load distribution [23]. As heat integration measures require modifications in the heat exchangers, HEN synthesis was also included in some of the methods [24]. Despite the fact that inter-plant heat transfer directly with process streams is considered impractical in most studies, some methods available in the literature still considered it as an option. Zhang et al. [25] introduced a HEN optimisation method for hot direct discharges/feeds between plants. A larger heat recovery was achieved by using process streams directly instead of intermediate fluids; however, issues regarding the implementation of such exchanges were not addressed. Direct heat exchange between processes requires more piping than using an intermediate fluid and hence a higher piping investment cost. Wang et al., studied the heat integration of direct, indirect and combined methods of multiple plants [26]. They concluded that direct exchange is most beneficial method for short distances while combined methods are best for medium distances and indirect transfer should be used for long distances. However, the conclusion was case-dependent and could not be generalised.

The main focus in inter-plant heat integration is excess heat recovery between plants. Since different processes have different pinch temperatures, the excess heat of one plant can be useful for another one. Based on this phenomenon, Rodera and Bagajewicz developed a method for optimal

integration of intermediate fluids in inter-plant heat transfer [27]. First, the targets for inter-plant exchange were identified using linear programming (LP) and source and sink plants were determined. Then, the optimal placement of the intermediate fluid circuit was identified using a mixed integer linear programming (MILP) formulation. Afterwards, the method was extended from two plants to n-plants [28]. When plants with similar pinch points are considered, recovering heat between them using an intermediate fluid might not be feasible. Building on previous work of Reference [28], Bagajewicz and Barbaro developed a method which uses HPs to upgrade the temperature of the excess heat from one plant and use it elsewhere [29].

Stijepovic and Linke also worked on optimal heat recovery in industrial zones focusing on excess heat [30]. They identified the excess heat potential of the plants manually and calculated the maximum heat recovery potential using LP. Finally the optimal heat recovery network was found using a mixed integer non-linear programming formulation. However, intra-plant process integration and improvements through more efficient energy conversion technologies were not included in the method. The layout constraints or location aspects were considered directly or indirectly in several methods. Kantor et al. [31] formulated the problem as a set of nodes and connections between them. The location aspects were included by adding the cost of resource transportation. Transportation methods were defined for each material sharing potential and an appropriate method for each was established as a result of the optimisation. Becker and Maréchal [32] proposed a MILP method to divide the system into smaller subsystems based on their locations. The subsystems were allowed to exchange heat only using heat transfer systems represented by intermediate fluids. This way, direct heat exchange over long distances was prevented. Pouransari and Maréchal [33] extended the previous problem to a heat load distribution (HLD) formulation. Implementation of sub-systems helped solving large-scale HLD problems, which are often computationally expensive. Bade and Bandyopadhyay [34] worked on a method to minimise the flow of a hot oil circuit between two plants. Although pumping and piping costs were not considered in the objective function, they were indirectly minimised by selecting the lowest possible hot oil flowrate.

HEN synthesis is a difficult problem to solve even for single plants [35]. When multiple plants and inter-plant heat integration are considered, it becomes even more challenging to obtain convergence. Song et al., combined the strengths of PA and MP in their work. In the first step, they divided the problem into smaller sections using an algorithm based on PA [36]. Then they carried out HEN synthesis of each section and finally optimised the inter-plant flows taking into account the pumping and piping costs [37]. Chang et al. [38] also proposed a method to simultaneously optimise the HEN and heat integration between plants. To simplify the problem, they considered a case with only two plants and using only a hot water loop to realise the heat exchange between them. The method was subsequently extended to more than two plants using different options (e.g., steam, hot oil) as intermediate fluids [39]. When a HEN is designed for more than one plant, it is important to determine the locations of the heat exchangers. Nair et al. [40] developed a MINLP method taking into account the locations of the heat exchangers in an eco-industrial park. They assumed that the temperature difference in the heat streams is linearly correlated to the travelled distance. They also considered piping and pumping costs and their trade-off with the operating cost benefits of heat recovery. Kachacha et al. [41] also considered the impact of plant location in the HEN problem by including piping and pumping costs. However, in order to keep the formulation linear, they made simplifying assumptions by using pre-calculated logarithmic mean temperature difference (LMTD) and pipe diameters. Laukkanen and Seppala [42] studied using nanofluids in inter-plant HEN synthesis. They developed a method to optimise the HEN, taking into account the trade-off between enhanced heat transfer and increased pumping power requirement due to the addition of nano-particles in the heat transfer fluid. Liu et al. [43] combined the efforts in mass integration and HEN synthesis in their heat integrated water allocation network model. Although they considered piping requirements for the water streams, they ignored heat losses and pumping requirements for transferring heat between the plants.

The literature of PI is rich in methods focused on inter-plant exchanges; however, graphical methods often neglect aspects related to plant layout. The most elaborate PA-based methods consider only heat integration using intermediate fluids and calculate heat losses and piping after integration. The MP methods address the location-based issues in inter-plant exchanges more extensively. However, most of the methods simplify the problem by considering the exchange only between two plants [38], identifying the excess heat manually [30] and optimising its valorisation instead of the overall system. Moreover, the integration of new utility systems was not a part of the optimisation [41], which might cause energy efficiency improvement opportunities to be missed. Another aspect overlooked in the literature is the type of the intermediate fluid which is used in inter-plant exchange. Methods have been specifically developed for heat sharing by steam [11], hot water [38] or hot oil [34]. Thus, the gaps in the literature are identified as:

1.  not considering the simultaneous integration of energy conversion technologies and inter-plant heat and material exchange infrastructure,
2.  only partially accounting for location aspects and
3.  case-specific methods and lack of generalised applicability.

The work presented in this paper addresses such gaps by formulating the utility integration problem taking into account the location aspects. The method is generic and offers flexibility in integration of new technologies as well as infrastructure for inter-plant heat and material exchange and carries out their optimisation simultaneously.

## 3. Method

The method proposed in this work is a novel MILP formulation based on [44]. The unique aspect that this work introduces is the location of plants and distance between them to understand their impact on PI targets.

### 3.1. Definition of Main Sets

The basis of the method is modeling a system with mass and energy balances. The following mains sets are therefore defined:

*   **TT**: The set of time steps to include the time-dependency of the system;
*   **LC**: The set of plant locations in the optimisation problem;
*   **U**: The set of units. Units are entities that represent an equipment (e.g., distillation column), a production unit or an entire plant, depending on its boundaries;
*   **S**: The set of streams. Streams represent flows of energy or materials;
*   **H**: The set of heat streams $\mathbf{H} \subset \mathbf{S}$. This set is used to create hot and cold streams;
*   **Z**: The set of resource streams $\mathbf{Z} \subset \mathbf{S}$. This set is used to model material flows (e.g., water, natural gas);
*   **ES**: The set of electricity streams $\mathbf{ES} \subset \mathbf{S}$. This set is used to model the electricity flows;
*   **L**: The set of layers. Resource streams are assigned to a member (i.e., layer) in this set where the material is specified;

The problem is defined as a system which has time-dependent behaviour, to be able to capture different operating modes of plants in different time steps ($t \in \mathbf{TT}$). The locations ($lc \in \mathbf{LC}$) divide the system into smaller subsystems, in which the mass and energy balances are closed. The locations can exchange heat and resources with each other using only selected streams (i.e., inter-location streams) while electricity flows freely in the system, without location restrictions. In each location, there can be one or more units ($u \in \mathbf{U}$) that are characterised by streams ($s \in \mathbf{S}$). Streams can represent heat deficit (e.g., cold streams), heat excess (e.g., hot streams), resource deficit/excess and electricity deficit/excess.

Figure 1 depicts an illustrative example summarising the method. When a heat stream is used within a location it is not subject to heat losses. Conversely, heat losses apply to use the heat in

another location. Streams ($s \in \mathbf{S}$) which are used in a location different from their original one have a pressure drop associated to the transfer and a corresponding pumping requirement. The necessary infrastructure (i.e., piping) which must be installed to realise inter-location exchange is also considered in this work.

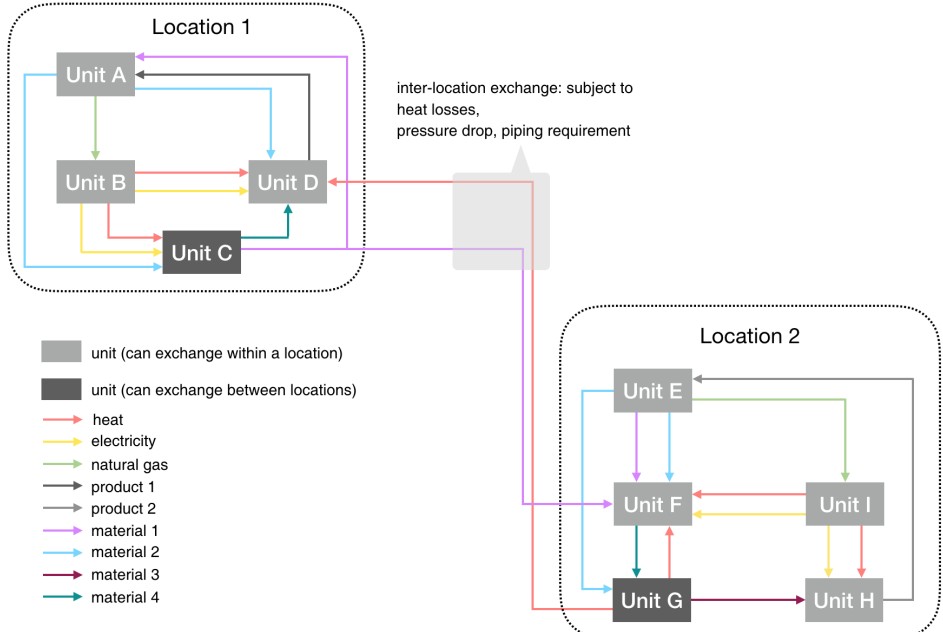

**Figure 1.** Simple graphical representation of the method.

### 3.2. Objective Function

Parametric optimisation is carried out with multiple objectives to generate and evaluate several scenarios. In MILP techniques used in PI, several objective functions are available in the literature. Economic objectives are the most applicable for this work, since the impact of distance can be monetised. Thus, the main objective (Equation (1)) is selected as the overall cost of the system excluding the investment in pipes for inter-location connections. The piping cost is selected as the second objective and integrated in the MILP framework as an $\epsilon$-constraint (see Equation (2)).

$$\min \quad C^{op} + C^{inv}, \tag{1}$$

$$C^{pipe} \leq \epsilon. \tag{2}$$

$C^{op}$ represents the operating cost associated with the consumption of resources (Equation (3)) while $C^{inv}$ is the annual investment cost for integrating new energy conversion technologies (Equation (4)) and $C^{pipe}$ is the annualised investment cost for piping between locations.

$$C^{op} = \sum_{u \in \mathbf{U}} \left[ \sum_{t \in \mathbf{TT}} \left( c_u^{op1} \cdot y'_{u,t} + c_u^{op2} \cdot f'_{u,t} \right) \cdot \Delta t_t^{op} \right], \tag{3}$$

$$C^{inv} = \left[ \sum_{u \in \mathbf{U}} \left( c_u^{inv1} \cdot y_u + c_u^{inv2} \cdot f_u \right) \right] \cdot F^{an}, \tag{4}$$

where $c_u^{op1}$ and $c_u^{inv1}$ are the fixed operating and investment costs of the units associated with their activation, $c_u^{op2}$ and $c_u^{inv2}$ are the variable operating and investment costs of the units which depend on their size, $\Delta t_t^{op}$ is the operating time and $F^{an}$ is the annualisation factor based on interest rate and lifetime of the equipment.

### 3.3. Sizing and Scheduling

Sizing and scheduling constraints determine if units are used in a certain time step (i.e., scheduling) as well as the purchased capacity and the utilised capacity in each (i.e., sizing). The units can be divided in two categories based on their behaviour: process units ($pu \in$ **PU** $\subset$ **U**) and utility units ($uu \in$ **UU** $\subset$ **U**). Process units have fixed size and scheduling and represent the production units on industrial plants. The sizing and scheduling constraints for the process units are defined in Equations (5)–(8).

$$f'_{u,t} = 1 \quad \forall\, u \in \textbf{PU},\, t \in \textbf{TT}, \tag{5}$$

$$f_u = 1 \quad \forall\, u \in \textbf{PU}, \tag{6}$$

$$y'_{u,t} = 1 \quad \forall\, u \in \textbf{PU},\, t \in \textbf{TT}, \tag{7}$$

$$y_u = 1 \quad \forall\, u \in \textbf{PU}, \tag{8}$$

where $f_u$ and $f'_{u,t}$ are the overall sizing factor and the sizing factor at time step $t \in$ **TT** and $y_u$ and $y'_{u,t}$ are binary variables which decide if a unit is purchased and utilised in time step $t \in$ **TT**, respectively. Hence although decision variables are defined for process units, they are eliminated by fixing their values.

Utility units are, defined with a certain size (i.e., reference size) but can be used in smaller and larger sizes as they scale with the sizing factor ($f$) according to the requirements of the process units. The equations governing the sizing and scheduling of utility units are Equations (9)–(12).

$$F_u^{min} \cdot y_u \leq f_u \leq F_u^{max} \cdot y_u \quad \forall\, u \in \textbf{UU}, \tag{9}$$

$$f'_{u,t} \leq f_u \quad \forall\, u \in \textbf{UU},\, t \in \textbf{TT}, \tag{10}$$

$$F_u^{min} \cdot y'_{u,t} \leq f'_{u,t} \leq F_u^{max} \cdot y'_{u,t} \quad \forall\, u \in \textbf{UU}, \tag{11}$$

$$y'_{u,t} \leq y_u \quad \forall\, u \in \textbf{UU},\, t \in \textbf{TT}, \tag{12}$$

where $F_u^{min}$ and $F_u^{max}$ are the lower and upper bounds of the sizing factor $f_u$ respectively. More detail and explanations on the sizing and scheduling constraints can be found in [44].

### 3.4. Resource Balance and Links

The resource balance is closed for each layer in the overall system as well as in each location. Moreover, resource links are included to observe and limit the flow of resources between units. The following sets are defined for the resource balance constraints:

- $\textbf{Z}_{u,l}$: This set consists of resource streams on layer $l \in$ **L** in unit $u \in$ **U**;
- $\textbf{U}_l$: The set of units of layer. This set consists of the units which have at least one resource stream on layer $l \in$ **L**;
- $\textbf{U}_{lc}$: The set of units of location. This set comprises of the units in a given location $lc \in$ **LC**;
- $\textbf{U}_{l,lc}$: The set of units of layer and location. This set includes the units in location $lc \in$ **LC**, which have at least one resource stream on layer $l \in$ **L** $\therefore$ $\textbf{U}_l \cap \textbf{U}_{lc}$;
- $\textbf{OL}_{lc}$: The set of other locations. For a given location $lc \in$ **LC** this set includes all the other locations in the system $\therefore$ $o \in \textbf{OL}_{lc} : o \neq lc$;
- $\textbf{OL}_u$: The set of other locations of a unit. For a given unit $u \in$ **U** this set contains all the locations except for the original location of the unit. It is specifically useful for units which can transfer flows to other locations;
- $\textbf{RL}_{l,u}$: The set of resource links of a unit. For a given layer $l \in$ **L** and a unit of that layer $u \in \textbf{U}_l$ this set consists of the other units on the same layer $\therefore$ $i \in \textbf{RL}_u : i \neq u$;

A unit can have several resource streams in the same layer; however, when the interactions of the units are considered, the flows in the same layer should be aggregated. This is enforced by Equation (13).

$$\dot{m}_{l,u,t}^{in} = \sum_{z \in \mathbf{Z}_{u,l}} \dot{m}_{l,z,t}^{in}, \quad \dot{m}_{l,u,t}^{out} = \sum_{z \in \mathbf{Z}_{u,l}} \dot{m}_{l,z,t}^{out} \quad \forall\, l \in \mathbf{L},\, u \in \mathbf{U}_l,\, t \in \mathbf{TT}, \tag{13}$$

where $\dot{m}_{l,u,t}^{in}$ and $\dot{m}_{l,u,t}^{out}$ are the in/out reference flows of unit $u \in \mathbf{U}$ and $\dot{m}_{l,r,t}^{in}$ and $\dot{m}_{l,r,t}^{out}$ are the inlet and outlet reference resource stream flows, respectively. Since the unit sizes vary depending on the scaling factor ($f$), the flows should also be scaled with respect to their units (see Equation (14)).

$$\dot{M}_{l,u,t}^{in} = \dot{m}_{l,u,t}^{in} \cdot f_{u,t}', \quad \dot{M}_{l,u,t}^{out} = \dot{m}_{l,u,t}^{out} \cdot f_{u,t}' \quad \forall\, l \in \mathbf{L},\, u \in \mathbf{U}_l,\, t \in \mathbf{TT}, \tag{14}$$

where $\dot{M}_{l,u,t}^{in}$ and $\dot{M}_{l,u,t}^{out}$ are scaled flows into/out of the unit, respectively. The overall resource balance (see Equation (15)) is included such that resource requirements of the units in the system are fulfilled by the other units.

$$\sum_{u \in \mathbf{U}_l} \dot{M}_{l,u,t}^{in} = \sum_{u \in \mathbf{U}_l} \dot{M}_{l,u,t}^{out} \quad \forall\, l \in \mathbf{L},\, t \in \mathbf{TT}. \tag{15}$$

The resource flow from each unit $\mathbf{U}_l$ is transferred to the other units $i \in \mathbf{RL}_{l,u}$ in the system via resource links. Similarly, the total resource flow into a unit is the sum of the flows from the units $j \in \mathbf{RL}_{l,u}$ in the resource links. These two conditions are combined in a single constraint (Equation (16)). The resource flow from a unit to the others is limited to its outflow. This constraint is imposed by Equation (17)

$$\dot{M}_{l,u,t}^{out} + \sum_{i \in \mathbf{RL}_{l,u}} \dot{M}_{l,i,u,t}^{rl} = \dot{M}_{l,u,t}^{in} + \sum_{j \in \mathbf{RL}_{l,u}} \dot{M}_{l,u,j,t}^{rl} \quad \forall\, l \in \mathbf{L},\, u \in \mathbf{U}_l,\, t \in \mathbf{TT}, \tag{16}$$

$$\sum_{j \in \mathbf{RL}_{l,u}} \dot{M}_{l,u,j,t}^{rl} \leq \dot{M}_{l,u,t}^{out} \quad \forall\, l \in \mathbf{L},\, u \in \mathbf{U}_l,\, t \in \mathbf{TT}, \tag{17}$$

where $\dot{M}_{l,i,u,t}^{rl}$ and $\dot{M}_{l,u,j,t}^{rl}$ are positive continuous variables which represent the flows in layer $l \in \mathbf{L}$ from unit $i \in \mathbf{RL}_u$ to $u \in \mathbf{U}_l$ and from $u \in \mathbf{U}_l$ to $j \in \mathbf{RL}_u$, respectively. Some units can exchange resources only within their origin location while others can have inter-location resource transfer. For units with inter-location resource exchange, a split factor is defined to determine the magnitude of the resource flow transferred to the other locations as seen in Equation (18).

$$\sum_{i \in \mathbf{U}_{l,o}} \dot{M}_{l,i,u,t}^{rl} = \dot{m}_{l,u,t}^{out} \cdot a_{l,lc,o,u,t} \quad \forall\, l \in \mathbf{L},\, lc \in \mathbf{LC},\, o \in \mathbf{OL}_{lc},\, u \in \mathbf{U}_l,\, t \in \mathbf{TT}, \tag{18}$$

where $a_{l,lc,o,u,t}$ is the split factor of the flow in layer $l \in L$ in unit $u \in \mathbf{U}_l$ from location $lc \in \mathbf{LC}$ to the other locations $o \in \mathbf{OL}_{lc}$. The resource balance is closed within locations as well as for the overall system. For a given location $lc \in \mathbf{LC}$ and layer $l \in \mathbf{L}$, the supply flows are those from the units of that location $u \in \mathbf{U}_{l,lc}$ and inter-location flows from the other locations. Similarly, the demand flows are those to the units of that location $u \in \mathbf{U}_{l,lc}$ and the inter-location flows to the other locations. Equation (19) ensures that the supply flow in a location is equal to the demand.

$$\begin{aligned}
&\sum_{o \in \mathbf{OL}_{lc}} \sum_{u \in \mathbf{U}_{l,o}} \dot{m}_{l,u,t}^{out} \cdot a_{l,o,lc,u,t} + \sum_{u \in \mathbf{U}_{l,lc}} \dot{M}_{l,u,t}^{out} \\
&= \sum_{o \in \mathbf{OL}_{lc}} \sum_{u \in \mathbf{U}_{l,o}} \dot{m}_{l,u,t}^{out} \cdot a_{l,lc,o,u,t} + \sum_{u \in \mathbf{U}_{l,lc}} \dot{M}_{l,u,t}^{in} \quad \forall\, l \in \mathbf{L},\, lc \in \mathbf{LC},\, t \in \mathbf{TT}.
\end{aligned} \tag{19}$$

### 3.5. Distribution Heat Losses

Direct heat exchange between process streams has been considered as an option for heat recovery in several case studies [45]; however, its drawbacks have also been highlighted [11]. Especially when sharing heat over long distances, direct exchange between process streams becomes impractical. Therefore, the distribution of heat between locations is carried out by using intermediate heat transfer media (e.g., hot water, steam, etc.).

Conceptually, heat streams that are allowed to exchange between locations are first duplicated and assigned to the other locations in the system. All duplicates as well as the stream itself are then assigned to a stream parent. Stream parents, although possessing no physical correspondence, are entities used to group the streams with their duplicates in different locations. There are two options for the heat streams that could be shared between different locations; to use them in their original location or to use their duplicate in other locations. These actions can be taken mutually; a stream can be used in its original location while its duplicate is used in another location. When the heat streams are used in the location of origin, it is assumed that heat losses do not occur, while heat losses and temperature drop apply for their duplicates in other locations. This is illustrated in Figure 2 considering heat distribution by steam and hot water as examples.

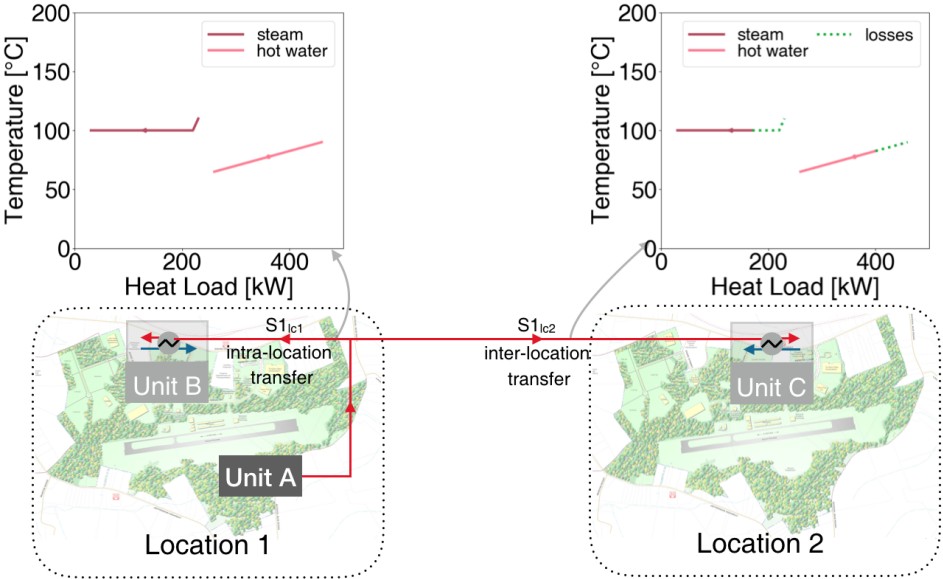

**Figure 2.** Distribution heat losses and their impact on the temperature-enthalpy profile.

In order to transfer heat between locations, pipes should be installed. To transport the fluid in urban areas, underground pipes are preferable because of regulations, but in other settings, above-ground pipes could be a better option since they have lower investment cost. However, as ground acts like an additional layer of insulation, heat losses in underground transfers are lower. Since both options are considered, another layer of decision is introduced in the problem, by creating duplicates of the streams for different transfer types (i.e., underground and above-ground). Heat losses in each transfer type are calculated following the methods explained in Sections 3.5.1 and 3.5.2. Considering the temperature drop associated with the heat losses, the temperature-heat profiles of the duplicate streams are reconstructed.

#### 3.5.1. Above-Ground Heat Losses

Heat losses in above-ground pipes occur because of heat transfer to ambient air. Assuming that the inner surface of the pipes is at the same temperature as the fluid flowing through them, the calculation

of the heat losses can be simplified. The overall heat transfer coefficient and the surface area of the heat distribution pipes are calculated using Equations (20) and (21).

$$\frac{1}{V_s} = \frac{1}{h^{\text{amb}}} + \frac{x_s^{\text{pipe}}}{\lambda_s^{\text{pipe}}} + \frac{x_s^{\text{ins}}}{\lambda_s^{\text{ins}}} \quad \forall\, s \in \mathbf{H}, \tag{20}$$

$$A_{s,lc,o}^{\text{pipe}} = 2 \cdot \pi \cdot d_s^{\text{ins}} \cdot l_{lc,o}^{\text{pipe}} \quad \forall\, s \in \mathbf{H},\ lc \in \mathbf{LC},\ o \in \mathbf{OL}_{lc}, \tag{21}$$

where $V_s$ is overall heat transfer coefficient, $h^{\text{amb}}$ is the convective heat transfer coefficient of ambient air, $\lambda_s^{\text{pipe}}$ and $\lambda_s^{\text{ins}}$ are the conductive heat transfer coefficients of pipe and insulation, $x_s^{\text{pipe}}$ and $x_s^{\text{ins}}$ are the thickness of the pipe and of the insulation material, $l_{lc,o}^{\text{pipe}}$, $d_s^{\text{ins}}$ and $A_{s,lc,o}^{\text{pipe}}$ are the length, insulated diameter and surface area of the pipe, respectively. The heat losses in the supply and return are then calculated using simple heat transfer equations (Equations (22) and (23)) and the remaining heat content of the stream is obtained by subtracting the distribution losses (Equation (24)).

$$\dot{q}_{s,lc,o,t}^{\text{sup}} = V_s \cdot A_{s,lc,o}^{\text{pipe}} \cdot \left( T_{s,t}^{\text{in}'} - T_t^{\text{amb}} \right) \quad \forall\, s \in \mathbf{H},\ lc \in \mathbf{LC},\ o \in \mathbf{OL}_{lc},\ t \in \mathbf{TT}, \tag{22}$$

$$\dot{q}_{s,lc,o,t}^{\text{ret}} = V_s \cdot A_{s,lc,o}^{\text{pipe}} \cdot \left( T_{s,t}^{\text{out}'} - T_t^{\text{amb}} \right) \quad \forall\, s \in \mathbf{H},\ lc \in \mathbf{LC},\ o \in \mathbf{OL}_{lc},\ t \in \mathbf{TT}, \tag{23}$$

$$\dot{q}_{s,t} = \dot{q}'_{s,t} - \dot{q}_{s,lc,o,t}^{\text{sup}} - \dot{q}_{s,lc,o,t}^{\text{ret}} \quad \forall\, s \in \mathbf{H},\ lc \in \mathbf{LC},\ o \in \mathbf{OL}_{lc},\ t \in \mathbf{TT}, \tag{24}$$

where $\dot{q}_{s,lc,o,t}^{\text{sup}}$ and $\dot{q}_{s,lc,o,t}^{\text{ret}}$ are the heat losses at supply and return, $T_{s,t}^{\text{in}'}$ and $T_{s,t}^{\text{out}'}$ are the inlet and the exit temperatures of the stream prior to heat losses, $\dot{q}'_{s,t}$ and $\dot{q}_{s,t}$ are the heat content of the stream prior to and after the heat losses and $T_t^{\text{amb}}$ is the ambient temperature.

3.5.2. Underground Heat Losses

The heat loss calculations for heat distribution with underground pipes is adapted from [46]. To simplify the problem, convection at the surface of the ground is converted to an equivalent layer of soil and added to the depth by Equation (25).

$$x^{\text{ground}} = x^{\text{ground}'} + \frac{\lambda^{\text{ground}}}{h^{\text{amb}}}, \tag{25}$$

where $\lambda^{\text{ground}}$ is the conductive heat transfer coefficient of ground, $x^{\text{ground}'}$ is the real pipe depth and $x^{\text{ground}}$ is the corrected pipe depth (i.e., thickness of soil). Heat losses also depend on the thermal resistance. In the case of heat distribution with double pipes (i.e., supply and return) the thermal resistance comes from the mutual interaction of the pipes, the ground and the insulation material. These parameters are calculated according to Equations (26)–(28).

$$X_s^{\text{mut}} = \frac{1}{4 \cdot \pi \cdot \lambda^{\text{ground}}} \cdot \ln \left[ 1 + \left( \frac{2 \cdot x^{\text{ground}}}{l^{\text{PP}}} \right)^2 \right] \quad \forall\, s \in \mathbf{H}, \tag{26}$$

$$X_s^{\text{ground}} = \frac{1}{2 \cdot \pi \cdot \lambda^{\text{ground}}} \cdot \ln \left( \frac{4 \cdot x^{\text{ground}}}{D_s^{\text{ins}}} \right) \quad \forall\, s \in \mathbf{H}, \tag{27}$$

$$X_s^{\text{ins}} = \frac{1}{2 \cdot \pi \cdot \lambda_s^{\text{ins}}} \cdot \ln \left( \frac{D_s^{\text{ins}}}{D_s^{\text{pipe}}} \right) \quad \forall\, s \in \mathbf{H}, \tag{28}$$

where $l^{\text{PP}}$ is the distance between the supply and return pipes, $D_s^{\text{pipe}}$ is the diameter of the pipe excluding the insulation and $X_s^{\text{mut}}$, $X_s^{\text{ground}}$ and $X_s^{\text{ins}}$ are the thermal resistances of the mutual

interaction between the pipes, ground and insulation material, respectively. The heat loss coefficients $W'_s$ and $W''_s$ are then calculated using Equation (29).

$$W'_s = \frac{X_s^{\text{ground}} + X_s^{\text{ins}}}{\left(X_s^{\text{ground}} + X_s^{\text{ins}}\right)^2 - X_s^{\text{mut}2}} \quad W''_s = \frac{X_s^{\text{mut}} + X_s^{\text{ins}}}{\left(X_s^{\text{ground}} + X_s^{\text{ins}}\right)^2 - X_s^{\text{mut}2}} \quad \forall s \in \mathbf{H}. \tag{29}$$

The heat losses at supply and return are calculated using Equations (30) and (31) and are subtracted from the heat load of the stream (see Equation (24))

$$\dot{q}_{s,lc,o,t}^{\text{sup}} = \left[\left(W'_s - W''_s\right) \cdot \left(T_{s,t}^{\text{in}} - T_t^{\text{ground}}\right) + W''_s \cdot \left(T_{s,t}^{\text{in}} - T_{s,t}^{\text{out}}\right)\right] \cdot l_{lc,o}^{\text{pipe}} \quad \forall s \in \mathbf{H}, \tag{30}$$
$$lc \in \mathbf{LC}, \; o \in \mathbf{OL}_{lc}, \; t \in \mathbf{TT},$$

$$\dot{q}_{s,lc,o,t}^{\text{ret}} = \left[\left(W'_s - W''_s\right) \cdot \left(T_{s,t}^{\text{out}} - T_t^{\text{amb}}\right) - W''_s \cdot \left(T_{s,t}^{\text{in}} - T_{s,t}^{\text{out}}\right)\right] \cdot l_{lc,o}^{\text{pipe}} \quad \forall s \in \mathbf{H}, \tag{31}$$
$$lc \in \mathbf{LC}, \; o \in \mathbf{OL}_{lc}, \; t \in \mathbf{TT}.$$

### 3.5.3. Modified Heat Cascade

The heat cascade constraints close the heat balance and make sure that heat flows from higher to lower temperatures. In the state-of-the-art targeting formulation [44], the heat balance is closed for the overall system. However, in this work with the introduction of locations, it is closed for each location, including the streams that can exchange between locations. The heat cascade set and parameter definitions are listed as follows:

- $\mathbf{K}_{lc}$: The set of temperature intervals of a location. An interval represents the zone above a certain temperature level ($T_k^{\text{lb}}$). Thus, this set is formed of intervals created by unique temperatures in each location in ascending order;
- $T_k^{\text{lb}}$: Temperature level of the interval $k \in \mathbf{K}_{lc}$. This parameter sets the lower bound of the interval;
- $\mathbf{HS}_{lc}$: The set of hot streams in each location ($lc \in \mathbf{LC}$). It includes the streams that are originally in the location as well as the inter-location streams from the other locations;
- $\mathbf{CS}_{lc}$: The set of cold streams in each location ($lc \in \mathbf{LC}$). It includes the streams that are originally in the location as well as the inter-location streams from the other locations;
- $\mathbf{HS}_{lc,k}$: The set of hot streams in each location ($lc \in \mathbf{LC}$) and temperature interval ($k \in \mathbf{K}$) $\therefore \mathbf{HS}_{lc,k} \in \mathbf{HS}_{lc}$. A hot stream $i \in \mathbf{HS}_{lc}$ is in a certain interval if its inlet temperature is higher than the temperature level of the interval $\therefore T_{i,t}^{\text{in}} \geq T_k^{\text{lb}}$
- $\mathbf{CS}_{lc,k}$: The set of cold streams in each location ($lc \in \mathbf{LC}$) and temperature intervals ($k \in \mathbf{K}$) $\therefore \mathbf{CS}_{lc,k} \in \mathbf{CS}_{lc}$. A cold stream $j \in \mathbf{CS}_{lc}$ is in a certain interval if its outlet temperature is higher than the temperature level of the interval $\therefore T_{j,t}^{\text{out}} \geq T_k^{\text{lb}}$
- $\mathbf{HI}$: The set of heat streams that are allowed to transfer heat between locations (i.e., inter-location streams) $\therefore \mathbf{HI} \subset \mathbf{H}$;
- $\mathbf{HI}_u$: The set of inter-location heat streams in unit $u \in \mathbf{U} \therefore \mathbf{HI}_u \subset \mathbf{HI}$
- $\mathbf{P}$: The set of heat stream parents. When a stream is allowed to be used in different locations (i.e., $\therefore s \in \mathbf{HI}$), it is duplicated in other locations as explained in Section 3.5. Parents are used to assign a stream and its duplicates to the same entity;
- $\mathbf{P}_u$: The set of stream parents of a unit $\therefore \mathbf{P}_u \subset \mathbf{P}$;
- $\mathbf{H}_u$: The set of heat streams of a unit $\therefore \mathbf{H}_u \subset \mathbf{H}$;
- $\mathbf{S}_p$: The set of streams of parents. Streams and their duplicates in other locations are aggregated in this set;
- $\mathbf{OL}_p$: Other locations of a parent. This set contains all the locations in the problem except for the original location of the parent;

- **TR**: The set of transfer types. The elements of this set are predefined as 'under-ground' and 'above-ground' since those are the transfer types considered for heat streams;
- $\mathbf{S}_{p,lc,tr}$: The set of streams of parents in each location and for each transfer type $\therefore \mathbf{S}_{p,lc,tr} \subset \mathbf{HI}$;

In order to be able to calculate the heat loads of the streams in the temperature intervals, their heat capacities are calculated according to Equation (32).

$$\dot{m}c_{Ps,t} = \frac{\dot{q}_{s,t}}{|T_{s,t}^{in} - T_{s,t}^{out}|} \quad \forall \, s \in \mathbf{H}, \, t \in \mathbf{TT}, \tag{32}$$

where $\dot{q}_{s,t}$ is the total reference heat load of the stream, $T_{s,t}^{in}$ is the inlet temperature, $T_{s,t}^{out}$ is the outlet temperature and $\dot{m}c_{Ps,t}$ is the heat capacity. In each interval, heat either flows from hot streams to the cold streams or is transferred to other intervals in the form of residual heat. This is enforced by Equation (33).

$$\left( \sum_{h \in \mathbf{HS}_{lc,k}} \dot{q}_{h,k,t} \cdot s_{h,t} \right) - \left( \sum_{c \in \mathbf{CS}_{lc,k}} \dot{q}_{c,k,t} \cdot s_{c,t} \right) - \dot{R}_{lc,k,t} = 0 \quad \forall \, lc \in \mathbf{LC}, \, k \in \mathbf{K}_{lc}, \, t \in \mathbf{TT}, \tag{33}$$

where $\dot{R}_{lc,k,t}$ is the continuous positive variable representing the residual heat in the interval, $\dot{q}_{h,k,t}$ and $\dot{q}_{c,k,t}$ are the reference heat loads of hot and cold streams in interval k, respectively, and $s_{h,t}$ and $s_{c,t}$ are scaling factors of the hot and cold streams, respectively. The reference load of a stream in an interval is equal to its total reference load if it is fully in the interval (Equations (34) and (35)). Otherwise the partial load of the stream in the interval is calculated (Equations (36) and (37)).

$$\dot{q}_{h,k,t} = \dot{m}c_{Ph,t} \cdot \left( T_{h,t}^{in} - T_{h,t}^{out} \right) \quad \forall \, h \in \mathbf{HS}_{lc,k}, \, lc \in \mathbf{LC}, \, k \in \mathbf{K}_{lc}, \, t \in \mathbf{TT}, \tag{34}$$

$$\dot{q}_{c,k,t} = \dot{m}c_{Pc,t} \cdot \left( T_{c,t}^{out} - T_{c,t}^{in} \right) \quad \forall \, c \in \mathbf{CS}_{lc,k}, \, lc \in \mathbf{LC}, \, k \in \mathbf{K}_{lc}, \, t \in \mathbf{TT}, \tag{35}$$

$$\dot{q}_{h,k,t} = \dot{m}c_{Ph,t} \cdot \left( T_{h,t}^{in} - T_k^1 \right) \quad \forall \, h \in \mathbf{HS}_{lc,k}, \, lc \in \mathbf{LC}, \, k \in \mathbf{K}_{lc}, \, t \in \mathbf{TT}, \tag{36}$$

$$\dot{q}_{c,k,t} = \dot{m}c_{Pc,t} \cdot \left( T_{c,t}^{out} - T_k^1 \right) \quad \forall \, c \in \mathbf{CS}_{lc,k}, \, lc \in \mathbf{LC}, \, k \in \mathbf{K}_{lc}, \, t \in \mathbf{TT}. \tag{37}$$

Abiding by the first law of thermodynamics, since energy cannot be created or destroyed, residual heat at the top and bottom intervals are set to zero (Equation (38)).

$$\dot{R}_{lc,k,t} = 0 \quad \forall \, lc \in \mathbf{LC}, t \in \mathbf{TT}, k = first(\mathbf{K}_{lc}) \text{ or } k = last(\mathbf{K}_{lc}). \tag{38}$$

The flow of a stream is scaled with its associated unit; hence, the scaling factor of a heat stream is equal to that of its unit (see Equation (39)). However, this applies only to the streams which cannot exchange between locations.

$$f'_{u,t} = s_{s,t} \quad \forall \, u \in \mathbf{U}, \, s \in \mathbf{H}_u, \, t \in \mathbf{TT} : \, s \notin \mathbf{HI}. \tag{39}$$

For the inter-location streams, splitting is taken into account using stream parents ($p \in \mathbf{P}$). A parent of an inter-location stream can be used in its original location as well as other locations. In addition, it can be transferred between locations using different transfer types ($tr \in \mathbf{TR}$). The sum of all splitting factors of a parent is equal to the scaling factor of its unit. This is enforced by Equation (40).

$$f'_{u,t} = \sum_{lc \in \mathbf{LC}} \sum_{tr \in \mathbf{TR}} b_{p,t,lc,tr} \quad \forall \, u \in \mathbf{U}, \, p \in \mathbf{P}_u, \, t \in \mathbf{TT}, \tag{40}$$

where $b_{p,t,lc,tr}$ is the splitting factor of parents in each location, time and for different transfer types. Similar to the relationship between units and streams, the streams of a parent scale together with the parent (Equation (41)).

$$s_{s,t} = b_{p,t,lc,tr} \quad \forall \, p \in \mathbf{P}, \, t \in \mathbf{TT}, \, lc \in \mathbf{LC}, \, tr \in \mathbf{TR}, \, s \in \mathbf{S}_{p,lc,tr}. \tag{41}$$

*3.6. Distribution Pump Work*

Heat and resource flows between locations are subject to pressure drop, which must be compensated by pumping. The pumping power requirement for inter-location exchange is considered by including additional electricity demand in the problem. The friction factor must be calculated first to estimate the pressure drop. Instead of the generic Colebrook equation, an explicit approximation by Haaland [47] is used in this work (see Equation (42)) as suggested by [48]. The Reynolds number is calculated using stream properties and pipe geometry, according to Equation (43).

$$\text{ff}_{s,t} = \left\{ -1.8 \cdot \log_{10} \left[ \left( \frac{\varepsilon_s}{3.7} \right)^{1.11} + \frac{6.9}{\text{Re}_{s,t}} \right] \right\}^{-2} \quad \forall \, s \in \mathbf{S}, \, t \in \mathbf{TT}, \tag{42}$$

$$\text{Re}_{s,t} = \frac{u_s \cdot d_s}{\nu_{s,t}} \quad \forall \, s \in \mathbf{S}, \, t \in \mathbf{TT}, \tag{43}$$

where $\nu_{s,t}$ is the kinematic viscosity, $u_s$ is the velocity, $\text{Re}_{s,t}$ is the Reynold's number, $\varepsilon_s$ is the pipe roughness and $\text{ff}_{s,t}$ is the friction factor. The pressure drop is then calculated using the Darcy-Weisbach equation, Equation (44).

$$\Delta P_{s,t,lc,o} = \text{ff}_{s,t} \cdot \left( \frac{l_{lc,o}^{\text{pipe}}}{d_s^{\text{pipe}}} \right) \cdot \left( \frac{\rho_{s,t}}{2} \right) \cdot u_{s,t}^2 \quad \forall \, s \in \mathbf{S}, \, t \in \mathbf{TT}, \, lc \in \mathbf{LC}, \, o \in \mathbf{OL}_{lc}, \tag{44}$$

where $\rho_{s,t}$ represents the density and $\Delta P_{s,t,lc,o}$ the pressure drop. Ignoring the mechanical inefficiencies of pumps, the required electricity to drive them is calculated by Equation (45).

$$\dot{e}_{s,t,lc,o}^{\text{pm}} = \Delta P_{s,t,lc,o} \cdot u_{s,t} \cdot \frac{\pi \cdot d_s^{\text{pipe}^2}}{4} \quad \forall \, s \in \mathbf{S}, \, t \in \mathbf{TT}, \, lc \in \mathbf{LC}, \, o \in \mathbf{OL}_{lc}, \tag{45}$$

where $\dot{e}_{s,t,lc,o}^{\text{pm}}$ is the reference pumping electricity requirements for the transfer of a stream between two locations. Resource streams require a transfer in only one direction since they are consumed at the target location. On the other hand, heat streams require bi-directional transfer because they have supply and return pipes. Thus, for heat streams, the pumping requirement is the sum of the electricity required in the supply and return pipes (see Equation (46)).

$$\dot{e}_{s,t,lc,o}^{\text{pm}} = \dot{e}_{s,t,lc,o}^{\text{sup}} + \dot{e}_{s,t,lc,o}^{\text{ret}} \quad \forall \, s \in \mathbf{H}, \, t \in \mathbf{TT}, \, lc \in \mathbf{LC}, \, o \in \mathbf{OL}_{lc}, \tag{46}$$

where $\dot{e}_{s,t,lc,o}^{\text{sup}}$ and $\dot{e}_{s,t,lc,o}^{\text{ret}}$ are the reference pumping power requirement at the supply and return respectively. It should be noted that the parameter $\dot{e}_{s,t,lc,o}^{\text{pm}}$ corresponds to a certain flow (e.g., reference flow). Hence, the pumping requirement of a flow scales with it. This is carried out for the resource (Equation (47)) and heat (Equation (48)) streams in two separate equations since different scaling factors are defined for different types of streams.

$$\dot{E}_{s,t,o}^{pm} = \dot{e}_{s,t,lc,o}^{\text{pm}} \cdot a_{l,lc,o,u,t} \quad \forall \, l \in \mathbf{L}, \, u \in \mathbf{U}_{lc}, \, t \in \mathbf{TT}, \, lc \in \mathbf{LC}, \, o \in \mathbf{OL}_{lc}, s \in \mathbf{Z}_{u,l}, \tag{47}$$

$$\dot{E}_{s,t,o}^{pm} = \dot{e}_{s,t,lc,o}^{\text{pm}} \cdot b_{p,t,o,tr} \quad \forall \, p \in \mathbf{P}, \, t \in \mathbf{TT}, \, lc \in \mathbf{LC}, \, o \in \mathbf{OL}_{lc}, \, s \in \mathbf{S}_{p,lc,tr}, \tag{48}$$

where $\dot{E}_{s,t,o}^{pm}$ is the pumping electricity requirement

### 3.7. Electricity Balance

Contrary to resource and heat streams, location restrictions do not apply to the electricity streams since the losses and infrastructure investment in the transfer of electricity are negligible compared to the others. To define the electricity balance, the following set is defined:

- $\mathbf{ES}_u$: This set consists of electricity streams in unit $u \in \mathbf{U}$;

Similar to resource streams, there might be several electricity streams in a unit. The reference flows of electricity in and out of a unit are calculated as the sum its electricity streams by Equation (49).

$$\dot{e}_{u,t}^{in} = \sum_{es \in \mathbf{ES}_u} \dot{e}_{es,t}^{in}, \quad \dot{e}_{u,t}^{out} = \sum_{es \in \mathbf{ES}_u} \dot{e}_{es,t}^{out} \quad \forall u \in \mathbf{U}, t \in \mathbf{TT}, \tag{49}$$

where $\dot{e}_{u,t}^{in}$ is the reference inflow and $\dot{e}_{u,t}^{out}$ is the reference outflow of electricity of unit $u \in \mathbf{U}$. The electricity flows of a unit, like other flows, scale with the unit itself. To obtain the real electricity generation/demand of a unit, the scaling factor is taken into account in Equations (50) and (51). In addition to the electricity streams, the demand of a unit includes the electricity for pumping resources and heat to other locations which is included in Equation (51).

$$\dot{E}_{u,t}^{out} = \dot{e}_{u,t}^{out} \cdot f_{u,t}' \quad \forall u \in \mathbf{U}, t \in \mathbf{TT}, \tag{50}$$

$$\dot{E}_{u,t}^{in} = \dot{e}_{u,t}^{in} \cdot f_{u,t}' + \left( \sum_{o \in \mathbf{OL}_{lc}} \sum_{s \in \mathbf{HI}_u} \dot{E}_{s,t,o}^{pm} \right) + \left( \sum_{o \in \mathbf{OL}_{lc}} \sum_{l \in \mathbf{L}} \sum_{s \in \mathbf{Z}_{u,l}} \dot{E}_{s,t,o}^{pm} \right) \quad \forall lc \in \mathbf{LC}, u \in \mathbf{U}_{lc}, t \in \mathbf{TT}, \tag{51}$$

where $\dot{E}_{u,t}^{out}$ and $\dot{E}_{u,t}^{in}$ are positive continuous variables representing the electricity supply and demand of the units, respectively. Finally, electricity demand of the units in the system is satisfied by the other units, which is imposed by Equation (52)

$$\sum_{u \in \mathbf{U}} \dot{E}_{u,t}^{in} = \sum_{u \in \mathbf{U}} \dot{E}_{u,t}^{out} \quad \forall t \in \mathbf{TT}. \tag{52}$$

### 3.8. Distribution Piping Cost

To realise heat and resource sharing between locations, the necessary infrastructure (i.e., pipeline) must be installed. Neglecting the investment for piping would result in unrealistically optimistic scenarios; therefore, this work includes the piping cost in the formulation. In multi-time problems, the flow of heat and resources between locations might vary in different time steps. The installed pipes must be capable of handling the flows at any time step; hence, the sizing of the pipes is carried out with respect to the maximum flow over all time steps (see Equations (53) and (54)).

$$\dot{Q}_{p,o,tr}^{max} = \max(\dot{q}_{s,t} \cdot b_{p,t,o,tr}) \quad \forall p \in \mathbf{P}, t \in \mathbf{TT}, s \in \mathbf{S}_p, o \in \mathbf{OL}_p, tr \in \mathbf{TR}_p, \tag{53}$$

$$\dot{M}_{l,o,u}^{max} = \max(\dot{m}_{l,u,t}^{out} \cdot a_{l,lc,o,u,t}) \quad \forall l \in \mathbf{L}, lc \in \mathbf{LC}, o \in \mathbf{OL}_{lc}, u \in \mathbf{U}_l, t \in \mathbf{TT}. \tag{54}$$

The max function is non-linear but can be converted to a set of linear constraints by introducing new continuous and binary variables. Linearisation of $\dot{Q}_{p,o,tr}^{max}$ is shown in Equations (55)–(60). A similar procedure is applied to linearise $\dot{M}_{l,o,u}^{max}$ but the equations are not included here.

$$\dot{q}_{s,t} \cdot b_{p,t,o,tr} \leq \dot{Q}_{p,o,tr}^{max} \quad \forall p \in \mathbf{P}, t \in \mathbf{TT}, s \in \mathbf{S}_p, o \in \mathbf{OL}_p, tr \in \mathbf{TR}, \tag{55}$$

$$\dot{G}_{p,t,o,tr} \leq \dot{q}_{s,t} \cdot b_{p,t,o,tr} \quad \forall p \in \mathbf{P}, t \in \mathbf{TT}, s \in \mathbf{S}_p, o \in \mathbf{OL}_p, tr \in \mathbf{TR}, \tag{56}$$

$$\dot{q}_{s,t} \cdot b_{p,t,o,tr} - \left( 1 - w_{p,t,o,tr} \right) \cdot \text{bigM} \leq \dot{G}_{p,t,o,tr} \quad \forall p \in \mathbf{P}, t \in \mathbf{TT}, s \in \mathbf{S}_p, o \in \mathbf{OL}_p, tr \in \mathbf{TR}, \tag{57}$$

$$\dot{Q}_{p,o,tr}^{max} \leq \dot{G}_{p,t,o,tr} + \left( 1 - w_{p,t,o,tr} \right) \cdot \text{bigM} \quad \forall p \in \mathbf{P}, t \in \mathbf{TT}, o \in \mathbf{OL}_p, tr \in \mathbf{TR}, \tag{58}$$

$$\dot{G}_{p,t,o,tr} \leq w_{p,t,o,tr} \cdot \text{bigM} \quad \forall \, p \in \mathbf{P}, \, t \in \mathbf{TT}, \, o \in \mathbf{OL}_p, \, tr \in \mathbf{TR}, \tag{59}$$

$$\sum_{t \in \mathbf{TT}} w_{p,t,o,tr} = 1 \quad \forall \, p \in \mathbf{P}, \, t \in \mathbf{TT}, \, o \in \mathbf{OL}_p, \, tr \in \mathbf{TR}, \tag{60}$$

where $\dot{Q}_{p,o,tr}^{max}$ is the maximum heat load of a stream parent per location and transfer type over time steps, $\dot{G}_{p,t,o,tr}$ is a slack variable which takes the value of $\dot{q}_{s,t} \cdot b_{p,t,o,tr}$ at the time step of the maximal load and 0 in the other time steps, $w_{p,t,o,tr}$ is a binary variable which takes the value of 1 at the time step of maximal load and 0 in other time steps and bigM is a large number for the big-M constraints [49].

Piping cost can be considered as a discrete function of the pipe diameter since there are standard pipe diameters and specific costs associated to them [50]. Such a relationship (see Table 1) is obtained by taking the average of the piping cost functions available in [51–54]. This relationship can be converted to flow-cost relationships using the stream properties.

**Table 1.** Piping cost for standard piping diameters.

| Standard pipe size | 1 | 2 | 3 | 4 | 5 | 6 | 7 | 8 | 9 | 10 | 11 | 12 |
|---|---|---|---|---|---|---|---|---|---|---|---|---|
| Diameter [mm] | 20 | 40 | 65 | 80 | 100 | 125 | 150 | 200 | 250 | 300 | 400 | 450 |
| Specific cost [€/m] | 96 | 166 | 250 | 312 | 387 | 480 | 580 | 775 | 975 | 1180 | 1588 | 1797 |

Each standard pipe size ($ps \in \mathbf{PS}$) has an upper bound ($\dot{q}_{p,o,tr,ps}^{ub}$ or $\dot{m}_{l,u,o,ps}^{ub}$) representing the maximum heat/mass that can flow through it as well as a lower bound ($\dot{q}_{p,o,tr,ps}^{lb}$ or $\dot{m}_{l,u,o,ps}^{lb}$). A new binary variable and a set of constraints are defined to determine the standard pipe size required for the flows. The constraints for the heat flow pipes are given in Equations (61)–(63).

$$\dot{q}_{p,o,tr,ps}^{lb} \cdot n_{p,o,tr,ps}^{h} \leq \dot{I}_{p,o,tr,ps}^{p} \leq \dot{q}_{p,o,tr,ps}^{ub} \cdot n_{p,o,tr,ps}^{h} \quad \forall \, p \in \mathbf{P}, \, o \in \mathbf{OL}_p, \, tr \in \mathbf{TR}, \, ps \in \mathbf{PS}, \tag{61}$$

$$\sum_{ps \in \mathbf{PS}} \dot{I}_{p,o,tr,ps}^{p} = \dot{Q}_{p,o,tr}^{max} \quad \forall \, p \in \mathbf{P}, \, o \in \mathbf{OL}_p, \, tr \in \mathbf{TR}, \, ps \in \mathbf{PS}, \tag{62}$$

$$\sum_{ps \in \mathbf{PS}} n_{p,o,tr,ps}^{h} \leq 1 \quad \forall \, p \in \mathbf{P}, o \in \mathbf{OL}_p, \, tr \in \mathbf{TR}, \, ps \in \mathbf{PS}, \tag{63}$$

where $n_{p,o,tr,ps}^{h}$ is a binary variable which takes the value of 1 if the heat flow is in the corresponding standard piping size and 0 otherwise and $\dot{I}_{p,o,tr,ps}^{p}$ is the slack variable which takes the value of $\dot{Q}_{p,o,tr}^{max}$ if $n_{p,o,tr,ps}^{h}$ is 1 and 0 otherwise. The sizes of the resource pipes are determined similar to the heat pipes according to Equations (64)–(66).

$$\dot{m}_{l,u,o,ps}^{lb} \cdot n_{l,u,o,ps}^{r} \leq \dot{J}_{l,u,o,ps} \leq \dot{m}_{l,u,o,ps}^{ub} \cdot n_{l,u,o,ps}^{r} \quad \forall \, l \in \mathbf{L}, \, u \in \mathbf{U}_l, \, o \in \mathbf{OL}_u, \, ps \in \mathbf{PS}, \tag{64}$$

$$\sum_{ps \in \mathbf{PS}} \dot{J}_{l,u,o,ps} = \dot{M}_{l,u,o,ps}^{max} \quad \forall \, l \in \mathbf{L}, \, u \in \mathbf{U}_l, \, o \in \mathbf{OL}_u, \tag{65}$$

$$\sum_{ps \in \mathbf{PS}} n_{l,u,o,ps}^{r} \leq 1 \quad \forall \, l \in \mathbf{L}, \, u \in \mathbf{U}_l, \, o \in \mathbf{OL}_u, \tag{66}$$

where $n_{l,u,o,ps}^{r}$ is a binary variable which takes the value of 1 if the resource flow uses the corresponding pipe size and 0 otherwise and $\dot{J}_{l,u,o,ps}$ is the slack variable which takes the value of $\dot{M}_{l,u,o,ps}^{max}$ if $n_{l,u,o,ps}^{r}$ is 1 and 0 otherwise.

After determining the piping sizes, their corresponding costs are calculated according to Equations (67) and (68), respectively. The total piping cost is then calculated by summing the heat and resource piping costs as presented in Equation (69).

$$C_{p,o,tr}^{pipe_h} = \sum_{ps \in \mathbf{PS}} c_{ps}^{pipe} \cdot tf \cdot l_{lc,o}^{pipe} \cdot n_{p,o,tr,ps}^{h} \quad \forall \, p \in \mathbf{P}, \, lc \in \mathbf{LC}, \, o \in \mathbf{OL}_{lc}, \, tr \in \mathbf{TR}, \tag{67}$$

$$C_{l,u,o}^{pipe_r} = \sum_{ps \in \mathbf{PS}} c_{ps}^{\text{pipe}} \cdot \text{tf} \cdot l_{lc,o}^{\text{pipe}} \cdot n_{l,u,o,ps}^{r} \quad \forall\ l \in \mathbf{L},\ u \in \mathbf{U}_l,\ lc \in \mathbf{LC},\ o \in \mathbf{OL}_{lc}, \tag{68}$$

$$C^{pipe} = \left( \sum_{p \in \mathbf{P}} \sum_{o \in \mathbf{OL}_p} \sum_{tr \in \mathbf{TR}} C_{p,o,tr}^{pipe_h} + \sum_{l \in \mathbf{L}} \sum_{u \in \mathbf{U}} \sum_{o \in \mathbf{OL}_u} C_{l,u,o}^{pipe_r} \right) \cdot \text{F}^{\text{an}} \\ \forall\ l \in \mathbf{L},\ u \in \mathbf{U}_l,\ lc \in \mathbf{LC},\ o \in \mathbf{OL}_{lc}, \tag{69}$$

where tf is the trenching cost factor (TCF) which is 1 for above-ground pipes (i.e., no trenching) and 1.3 for underground pipes [55], $c_{ps}^{\text{pipe}}$ is the specific piping cost of the corresponding pipe size and $C^{pipe}$ is the total annualised piping cost for heat and resource exchange between the locations.

## 4. Case Study

The case study consists of eight locations, each containing a plant and its associated utilities. The locations are assigned with x and y coordinates to indicate their geographical position. It is assumed that all locations have the same altitude; hence, the distance between locations is calculated according to the Manhattan distance (MD). The locations and their plants are given as follows:

- Site 1: Low-temperature chemicals production plant. The process requires heat for pre-heating the reactants and re-boiling the bottoms streams of distillation columns. The products separated at the distillation columns are cooled by air in overhead coolers first and then by water in shell and tube heat exchangers. Electricity requirement in the site is due to mechanical drives such as pumps and compressors. The site energy profile is adapted from [56];
- Site 2: Medium-temperature chemicals production plant. Similar to the plant at Site 1, this process requires heating by steam and cooling by water and air as well as electricity for the mechanical drives. The only difference is that this site has a higher pinch point and production rate. The site energy profile is adapted from [56];
- Site 3: Brewery plant. Brewery consists of two main processes; beer production and bottling. In beer production the raw materials are mixed, boiled, fermented and pasteurised. Bottling includes several stages of cleaning. The heating requirement is mainly in the brewhouse for heating and boiling the mixture prior to fermentation [57] and electricity demand is mainly due to refrigeration systems. The model of the site is adapted from [58];
- Site 4: Cement plant with dry process. The Cement process is centered around clinker production which requires heat at high temperatures up to (1450 °C) for preheating the raw materials to temperature required for calcination. After the reaction, the product at high temperature is cooled and milled to give its final form. The heating requirement is satisfied by burning coal and alternative fuels in the kiln while cooling is done by air. Electricity is required to drive the mills and other mechanical equipment. The process is modelled according to [59];
- Site 5: Dairy plant. Dairy plants include processes for multiple products. Depending on the product slate, the process characteristics differ significantly. For example, condensed milk production is heating intensive while ice cream production mostly requires refrigeration. The plant in the case study is assumed to produce condensed milk and yogurt. The model is adapted from [59];
- Site 6: Pulp and paper production plant. Pulp and paper plants may utilise different technologies for pulp production whereas paper production is relatively standard. The main energy consumption of the plant is heat for drying paper and pulp. The model of the plant is adapted from [60];
- Site 7: Oil refinery. Refineries rely on distillation to separate different components of crude oil, followed by several chemical reactions to break large hydrocarbon molecules into smaller ones. The core of the plant, as well as the main energy consumer, is the crude oil distillation unit. Similar to the chemical plants, heat is required for the bottom streams of the distillation columns and reactions, while cooling is required to condense the overhead streams of the distillation columns. The refinery model is adapted from [59];
- Site 8: Waste incineration plant. Waste incineration plants are typically located near cities to provide heat to the district heating networks while producing electricity in the steam network. The plant is modelled according to [7].

As the focus of the method is energy consumption, only the flows related to energy (e.g., fuels, heat, electricity) are taken into account. Thus the raw material and intermediate flows within each plant are neglected. The product flowrates are indicated to provide a reference size of the plants. The locations of the plants can be seen in Figure 3.

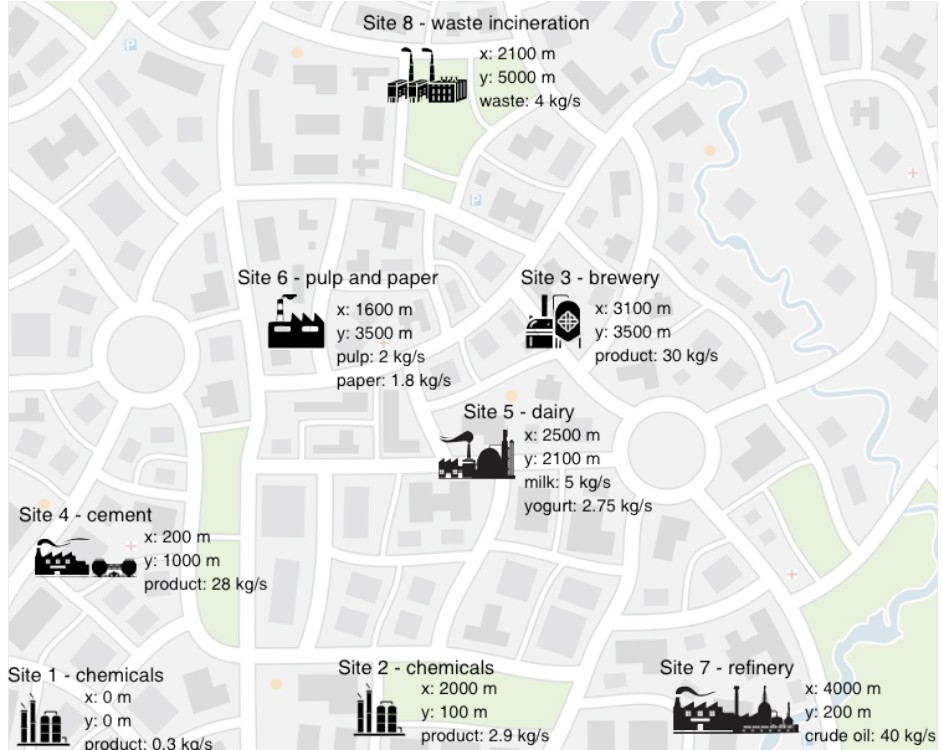

**Figure 3.** Case study layout.

The grand composite curves (GCCs) of the plants in Figure 4 give detailed information about the minimum heating and cooling requirements. The data used to construct the curves are given in the Appendix A.

### 4.1. Utility Systems and Resources

### 4.1.1. Existing Technologies on the Plants

Each plant in the system is considered to operate independently under current conditions. Thus, the sites have their own utility systems to close the energy balance and market access to close the resource balance. The utility systems that already exist on the sites are:

- Boiler: represents a combustion chamber which intakes natural gas and air and outputs heat. The boiler modelling is done according to [61] which assumes that the heat from natural gas consumption is delivered to the steam network by radiation and convection. Table 2 depicts the specifications of the boiler model;

**Table 2.** Boiler model specifications.

| Stream | $T^{in}$ [°C] | $T^{out}$ [°C] | $\dot{q}$ [kW] |
|---|---|---|---|
| Fuel | - | - | 1031 |
| Radiation | 827 | 827 | 656 |
| Convection | 827 | 100 | 324 |
| Air preheating | 25 | 150 | −49 |

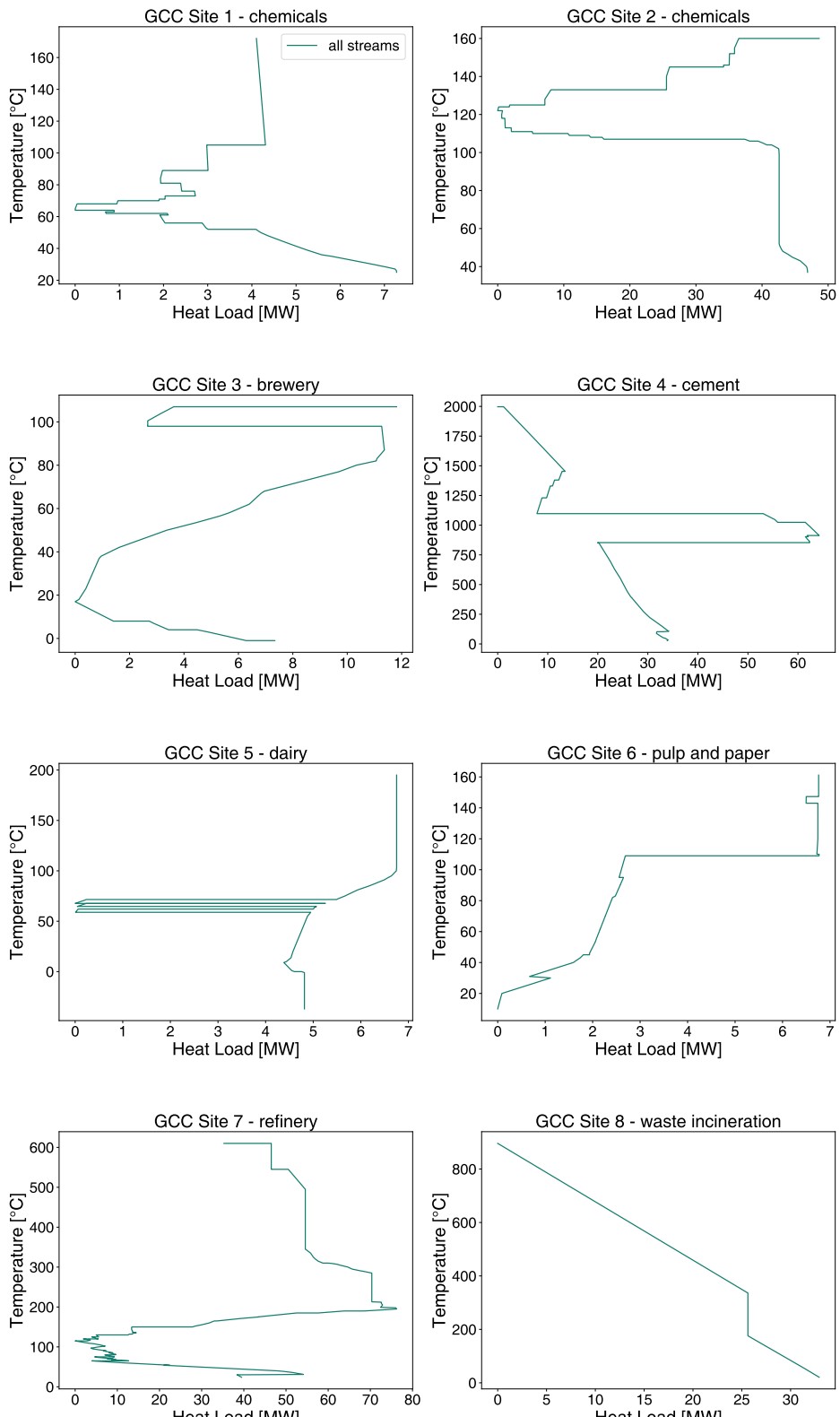

**Figure 4.** Grand composite curves (GCCs) of the processes in the case study.

- Steam network: an intermediate step between the boiler and processes. Heat is delivered to the steam network resulting in steam production. Afterwards, high pressure steam is turbined to cogenerate electricity and low pressure steam which is fed into the processes for heating.

The steam network model is adapted from [62]. Table 3 illustrates the configuration of the steam network in all locations except for Site 8. The configuration of the waste incineration steam network is adapted from [7];

**Table 3.** Configuration of the steam network.

| Type | Header Pressure [bara] | Header Temperature [°C] | Turbine |
|---|---|---|---|
| Production/Distribution | 45 | 367 | yes |
| Distribution | 24 | 228 | yes |
| Distribution | 8 | 175 | yes |
| Distribution | 4 | 150 | no |
| Production/Distribution | 2 | 126 | no |
| Production/Distribution | 1 | 105 | no |

- Cooling systems: consist of cooling towers and aerocoolers. Cooling water is used to remove heat from the processes which is then discharged to the environment at the cooling tower. The cooling tower model (see Table 4) is adapted from [63]. Aerocooling is modelled as a simple fan according to [56].

**Table 4.** Water cooling specifications.

| Stream | $T^{in}$ [°C] | $T^{out}$ [°C] | $\dot{q}$ [kW] | $\dot{m}$ [kg/s] | $\dot{e}$ [kW] |
|---|---|---|---|---|---|
| Supply | 15 | 25 | $-1000$ | - | - |
| Return | 25 | 15 | 1000 | - | - |
| Make-up water | - | - | - | 0.34 | - |
| Electricity | - | - | - | - | 10 |

- Refrigeration: is needed for the processes requiring sub-atmospheric temperatures (e.g., brewery, dairy). A simple refrigeration cycle model (see Table 5) is used which provides the process-specific refrigeration temperature and assuming a coefficient of performance of 3 [7].

**Table 5.** Refrigeration cycle specifications.

| Stream | $T^{in}$ [°C] | $T^{out}$ [°C] | $\dot{q}$ [kW] | $\dot{e}$ [kW] |
|---|---|---|---|---|
| Evaporation | $-5$ | $-5$ | $-1000$ | - |
| Condensation | 35 | 35 | 1333 | - |
| Electricity | - | - | - | 333 |

4.1.2. Additional Technologies for Improvements

In addition to the existing utility systems, new energy conversion technologies can be purchased and installed to improve the overall energy efficiency and operating cost. The additional utilities considered in the case study are:

- HPs: are ideal in the cases where the pinch temperature is low and heat transfer from below to above the pinch point with low temperature lift is possible. Considering the GCCs presented in Figure 4, Site 1, Site 2, Site 5 and Site 7 offer potentials for heat pump integration. The evaporation and condensation temperatures of the HPs are selected manually based on the GCCs. Table 6 depicts the specifications of the HPs;

**Table 6.** Heat pump specifications.

| | $T^{evap}$ [°C] | $T^{cond}$ [°C] | $\dot{q}^{evap}$ [kW] | $\dot{q}^{cond}$ [kW] | $\dot{e}^{comp}$ [kW] |
|---|---|---|---|---|---|
| $HP_{site1}$ | 58 | 73 | −1008 | 1067 | 59 |
| $HP_{site2}$ | 100 | 130 | −716 | 797 | 81 |
| $HP_{site5}$ | 82 | 107 | −1035 | 1131 | 96 |
| $HP_{site7}$ | 100 | 130 | −716 | 797 | 81 |

- Mechanical vapour recompressions (MVRs): similar to HP but instead of using an intermediate fluid, the vapour is directly compressed to a higher pressure and temperature. Potential MVR integration at Site 3 and Site 6 is considered. Steam can be imported at 1 bar from the other locations and compressed to 2 bar instead of producing it in the boilers.
- Cogeneration Engines: commonly used in industrial sites as they provide both heat and electricity. Similar to HP, pinch point plays an important role in the integration of cogeneration engines. A significant part of the heat comes from the cooling water of the engine. Thus, they are suitable only for the processes with low pinch temperatures such as Site 1, Site 3, Site 5 and Site 6. The specifications of the cogeneration engines (see Table 7) are adapted from [56].

**Table 7.** Cogeneration engine specifications.

| Stream | $T^{in}$ [°C] | $T^{out}$ [°C] | $\dot{q}$ [kW] | $\dot{e}$ [kW] |
|---|---|---|---|---|
| Fuel | - | - | 2605 | - |
| Exhaust Gasses | 470 | 120 | 537 | - |
| Engine Cooling | 87 | 80 | 653 | - |
| Electricity | - | - | - | 1063 |

- Photo-voltaics (PVs): have the potential to supply the electricity requirement of the industrial processes as well as the utility systems (e.g., HPs). The limitations for PV integration are availability of land and high capital investment. As the industrial plants are generally located outside urban centres, it is assumed that there is enough land and roof surface to install them. The PV model (see Table 8) is adapted from [7] for a reference area of 100 m$^2$.

**Table 8.** Photo-voltaic (PV) specifications.

| | $T^{in}$ [°C] | $T^{out}$ [°C] | $\dot{q}$ [kW] | $\dot{e}$ [kW] |
|---|---|---|---|---|
| PV | - | - | - | 16.6 |

### 4.1.3. Utility and Resource Costs

Energy conversion technologies not initially available on the sites require investment for the purchase and installation of the equipment. The HPs consist of two heat exchangers (i.e., evaporator and condenser) and a compressor while the MVRs require investment only in a compressor. The cost of the heat exchangers and compressors is calculated according to [64]. The non-linear cost functions are linearised within the range of application [$f_u^{min}$, $f_u^{max}$] to be coherent with the MILP framework. The fixed and variable investment cost ($c_u^{inv1}$ and $c_u^{inv2}$, respectively) of the additional technologies are listed in Table 9 and affect the objective function by inclusion in Equation (4).

The operating cost of the system is calculated based on the resource and energy consumption. Since raw materials and intermediate products are excluded from the analysis, only the costs of electricity, water and fuels are considered. Table 10 illustrates the specific cost of the main contributors to the operating cost. The specific cost [65], share in the fuel mix [66] and properties [67] of the alternative fuels used in the cement plant are given in the Appendix A.

**Table 9.** Costing and sizing parameters of the additional energy conversion technologies.

| Unit | $c^{inv1}$ [€/year] | $c^B$ [€/year] | $f^{min}$ [-] | $f^{max}$ [-] |
|---|---|---|---|---|
| $HP_{site1}$ | 8774 | 54,521 | 0.1 | 5 |
| $HP_{site2}$ | 5270 | 22,328 | 0.1 | 30 |
| $HP_{site5}$ | 8425 | 46,909 | 0.1 | 10 |
| $HP_{site7}$ | 5270 | 22,328 | 0.1 | 10 |
| $MVR_{site3}$ | 2265 | 26,950 | 0.1 | 4 |
| $MVR_{site6}$ | 4595 | 38,142 | 0.1 | 2 |
| $Co\text{-}gen_{site1}$ | 11,910 | 119,095 | 0.1 | 1 |
| $Co\text{-}gen_{site3}$ | 11,910 | 119,095 | 0.1 | 1 |
| $Co\text{-}gen_{site5}$ | 11,910 | 119,095 | 0.1 | 1 |
| $Co\text{-}gen_{site6}$ | 11,910 | 119,095 | 0.1 | 1 |
| PV | 0 | 70,730 | 0 | 100 |

**Table 10.** Specific cost of the main fuels and resources.

| Unit | $c^{spec}$ |
|---|---|
| Natural gas | 0.030 [€/kWh] |
| Coal | 0.600 [€/kg] |
| Electricity purchase | 0.092 [€/kWh] |
| Electricity selling | 0.055 [€/kWh] |
| Water | 0.070 [€/t] |

## 5. Results and Discussion

The method is first applied to two of the plants presented in the case study introduced in Section 4 to analyse the results in deeper detail. The optimisation of the overall case study is also completed to display the capability and effectiveness of the method in solving large-scale, complex industrial problems.

### 5.1. Symbiosis between Two Chemical Plants

The two chemical plants introduced in the case study represent an opportunity for industrial symbiosis by heat sharing. Site 2 has a higher pinch point than Site 1, thus its excess heat can be recovered and used in Site 1. Figure 5 illustrates the results of parametric optimisation in which the sum of the operating and utility investment cost of the plants is minimised while the investment in piping between the plants is constrained with a limit ($\epsilon$). Figure 5a exhibits the trade-off between the two objectives while the colour indicates the sum of both (i.e., total cost with piping). The minimum total cost with piping is obtained in Solution 6 when a cogeneration engine is installed on Site 1 and a heat pump on Site 2 while heat is shared between the sites with 1 bar steam; however, similar total cost results are obtained with other solutions where investments in piping are between 0 and 0.3 M€. As the investment cost limit for the pipeline between plants is increased, lower piping-exclusive total cost is obtained. This decrease coincides with lower operating cost due to excess heat recovery between the sites. Investment decisions also vary with respect to the limit on the piping investment. For example, investment on Site 1 HP decreases in correlation with piping investments because the imported steam from Site 2 replaces the HP. Conversely, integration of the cogeneration engine on Site 1 is independent of the piping allowance as the decision of installing the cogeneration engine is driven by the electricity generation rather than heat. PV is not integrated in any of the solutions as its capital investment cost is higher than the associated operating cost benefits. Figure 5b reveals further details about the solutions of parametric optimisation focusing on total investment cost including piping and heat sharing between the sites. When investment on piping between the sites is not allowed (Solution 1, piping cost = 0 M€/year), the optimal solution is investing in HPs on both sites and a cogeneration engine on Site 1. As the limit on piping investment increases, 1 bar steam exchange becomes part of the optimal solution. When 1 bar steam import reaches its limit (solution 6, piping cost = 0.29 M€/year),

the investments in exchanging 2 bar steam are made. When heat sharing and losses are considered, a significant decrease in the heat losses in the last solution (solution 10, piping cost = 0.53 M€/year) is observed, although the absolute heat exchanged remains the same as underground pipes are selected given the high allowance on piping cost. The same phenomenon can be observed comparing solutions 8 (piping cost = 0.41 M€/year) and 9 (piping cost = 0.45 M€/year). In the latter solution, more 2 bar steam is shared but heat losses are lower due to installation of underground pipes.

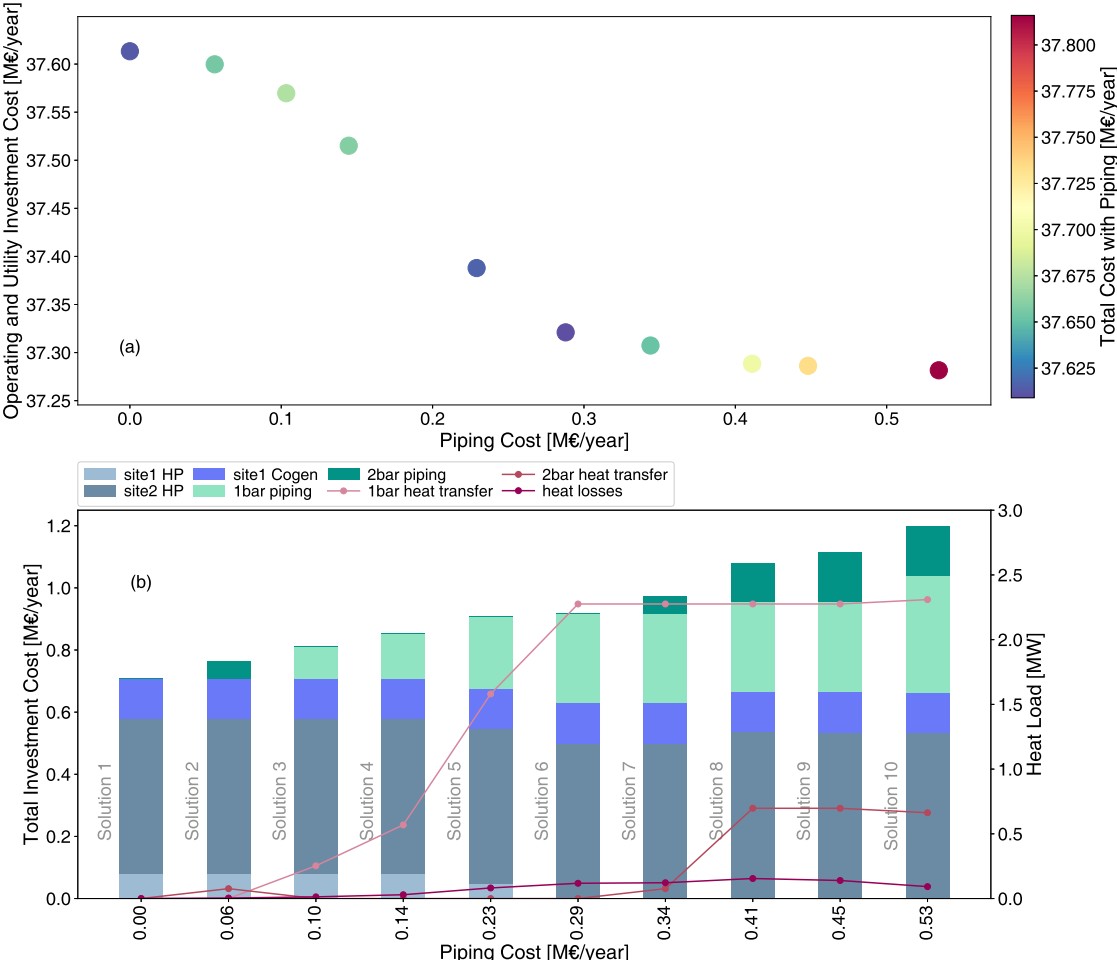

**Figure 5.** Parametric optimisation with two chemical plants. (**a**) pareto frontier; (**b**) investment cost breakdown and heat sharing.

Figure 6 depicts the Carnot composite curves (CCCs) of Site 1 and Site 2 which provide more detail about heat integration. HP at Site 1 is partially replaced by 1 bar steam import when solutions 1 and 4 are compared. In solution 7, 1 bar steam import from Site 2 completely replaces the HP at Site 1. Moreover, the size of the HP at Site 2 increases to produce 2 bar steam and export it to Site 1. In solution 10, the boiler at Site 1 is completely substituted by 1 and 2 bar steam imports.

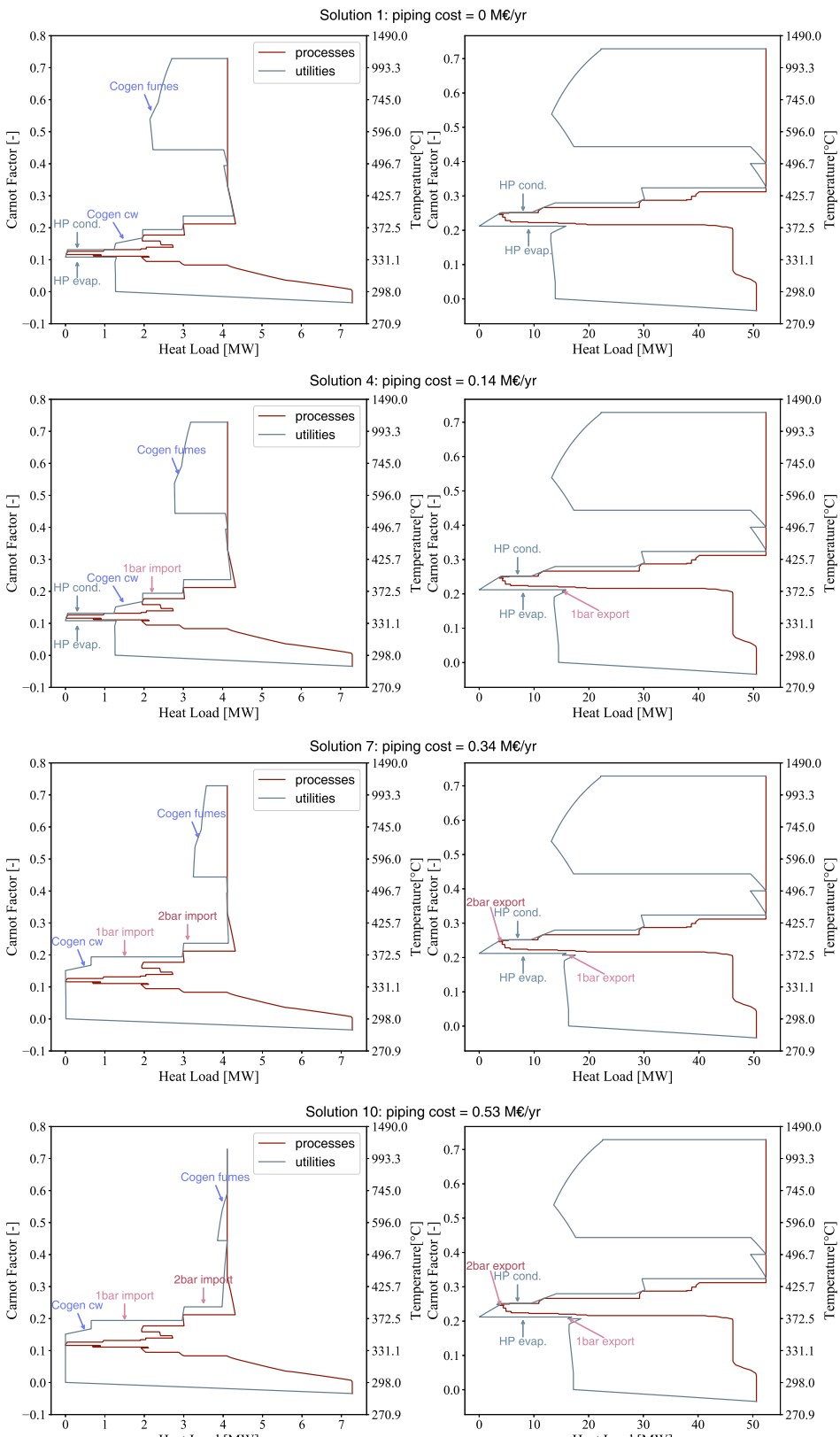

**Figure 6.** CCCs of Site 1 (**left**) and Site 2 (**right**) with different piping cost investment limits.

### 5.2. Symbiosis between All Plants

Optimisation of the case study introduced in Section 4 with all plants was completed to exhibit the capability of the method to solve complex, large-scale problems. Sites 2 and 7 have excess heat, while Sites 1, 3, 5 and 6 require heating at relatively low temperatures. In addition, the waste incineration plant at Site 8 can provide steam to the other locations instead of turbining and condensing it at low pressure. Thus, there is a large potential for heat sharing between the locations. The transfer of heat can be done only via utility systems. As there is a steam network in each location except Site 4, steam is considered to be the transfer fluid. In addition to symbiosis by sharing heat, there is a possibility to share material from Site 1 to Site 4, as the cement process can use chemical waste as a fuel. Similar to the case with two chemical plants, the total cost of the system excluding the piping cost is minimised, while constraining the piping investment cost at different limits.

Figure 7a depicts the parametric optimisation results. Similar to Section 5.1, the operating and utility investment cost decreases as the limit on piping investment is increased. Considering all cost elements, the minimum is obtained when the piping cost is ∼2.2 M€/year. Other solutions show that the operating and utility cost can be further decreased; however, the additional piping cost is larger than the associated economic benefit. A further ∼6 M€/year investment in piping to connect the sites only results in marginal improvement in the main objective. The breakdown of total investment cost can be seen in Figure 7b, as well as the amount of heat sharing and losses. In solutions with a low piping cost limit (similar in magnitude to the utility investment cost), all investments are relatively small. With increasing limits on the piping investment, it quickly dominates the investment cost distribution as heat sharing can be done over long distances and with high flowrates. At low piping investment limits, the investment goes toward sharing high pressure steam since it requires smaller pipe diameter. As the limit on the piping cost increases, heat sharing switches to lower pressure (i.e., lower temperature) steam as lower temperature corresponds to lower heat losses. With increasing allowance, heat losses trend upward until the pipe investment cost reaches ∼6.6 M€/year as more heat is shared between the sites. Past this level, despite stable or increased heat sharing, heat losses start decreasing since higher allowance on piping cost permits investment in underground pipes which are naturally more insulated.

Figure 8 illustrates the layout of the plants and their connections for four solutions with different piping investment levels from parametric optimisation (highlighted in Figure 7b. The switch from high to low pressure steam sharing can be observed by comparing solutions with lower (e.g., Figure 8a) and higher piping investments (e.g., Figure 8b). Sharing chemical waste with the cement plant is activated with a low piping investment allowance. Heat sharing is selected using above-ground pipes at low piping investment solutions (e.g., Figure 8a), while increasing this allowance first encourages a mixture of above-ground and underground pipes (e.g., Figure 8b,c) and then only underground pipes (e.g., Figure 8d). With large piping investment limits, even very small heat sharing options (<1 MW) are activated, which might not be practically feasible.

### 5.3. Piping Investment by a Third Party

Although investment in inter-plant infrastructure has proven economic benefits, industries are often reluctant to partake in such projects as they include several companies and the payback time may not fit within stringent economic policies. In such cases, the involvement of a third party could be considered. The analysis presented in Section 5.2 examines the piping investment to be made by industries but the same investment could alternatively be made by a third party with less stringent payback requirements. The investment to be made by such a third party would be recuperated by providing steam at a slight premium compared to the cost of generation. Such a strategy mitigates industrial risk and investment while providing improvements in operating cost and business opportunities for utility providers, which use different business models than the large process industries. The steam price provided to the industries is calculated to recuperate the piping cost over the time horizon of the installation and an additional premium to realise profitability for

the third party. The steam premium is varied between 0%—representing a non-profit third party or shared industrial investment—to 20%, providing a business case for the third party to compensate the investment and be profitable. The results are compared in Figure 9a based on the overall system profit. The baseline for the profitability analysis is the solution in which piping cost is zero, that is, there is no sharing between the industries and each plant pays for its own energy technologies. For the other solutions, the system profit is calculated as the difference between the change in operating cost and the annualised investment cost including piping. As the steam price premium increases, the profitable zone narrows and the highest potential profit decreases, since a higher price is paid for the same operating cost savings, due to the profit margin of the third party. The solutions with different steam price premiums selected in Figure 9a are compared in terms of steam price in Figure 9b. The price of steam is calculated as the ratio between the cost of piping including the profit margin of the third party and the total amount of steam shared between sites. The savings on steam price for the industries (compared to the break-even price) range between 47% and 32%. With increasing third-party profit, the steam price increases; however, it remains well below the break-even price.

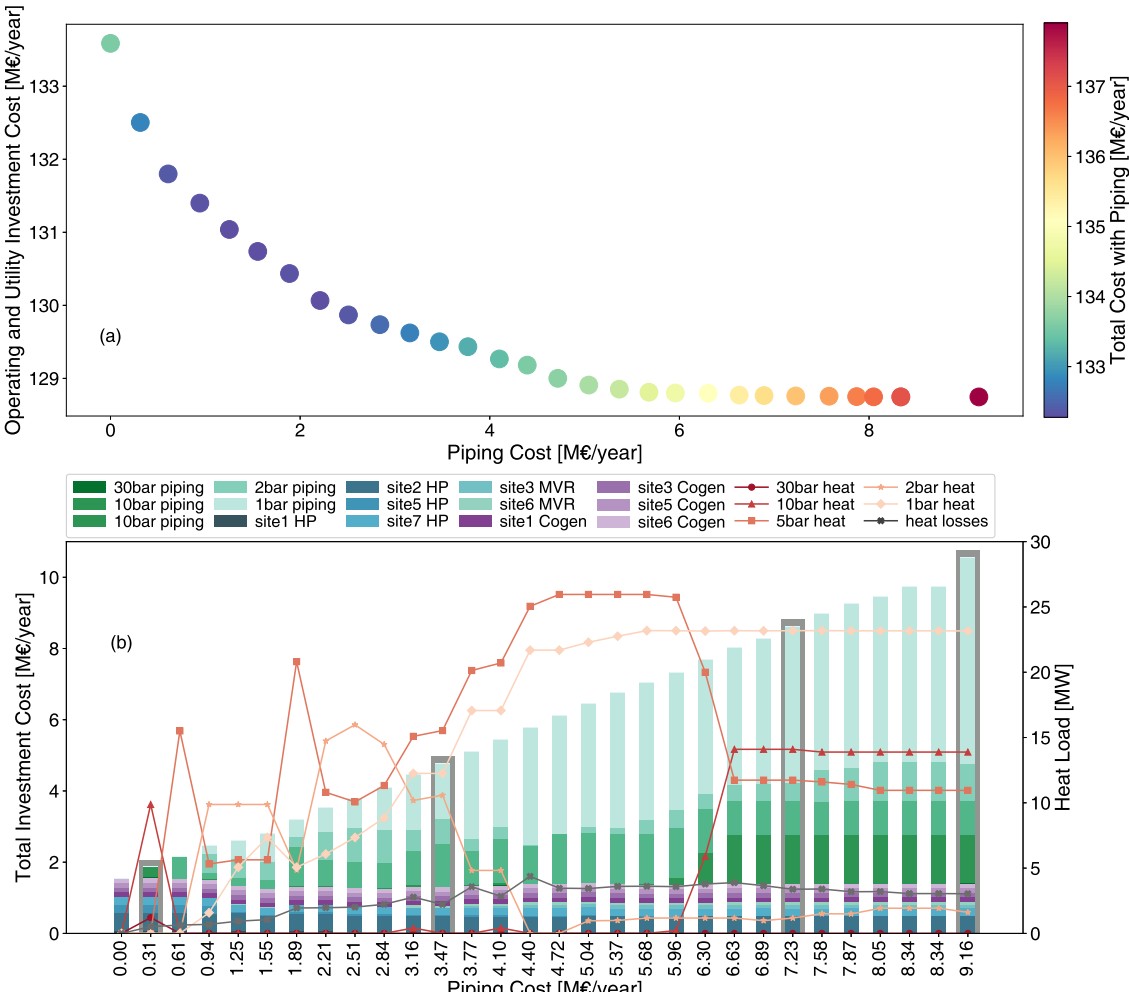

**Figure 7.** Parametric optimisation results for an industrial complex (**a**) pareto frontier; (**b**) investment cost breakdown and heat sharing.

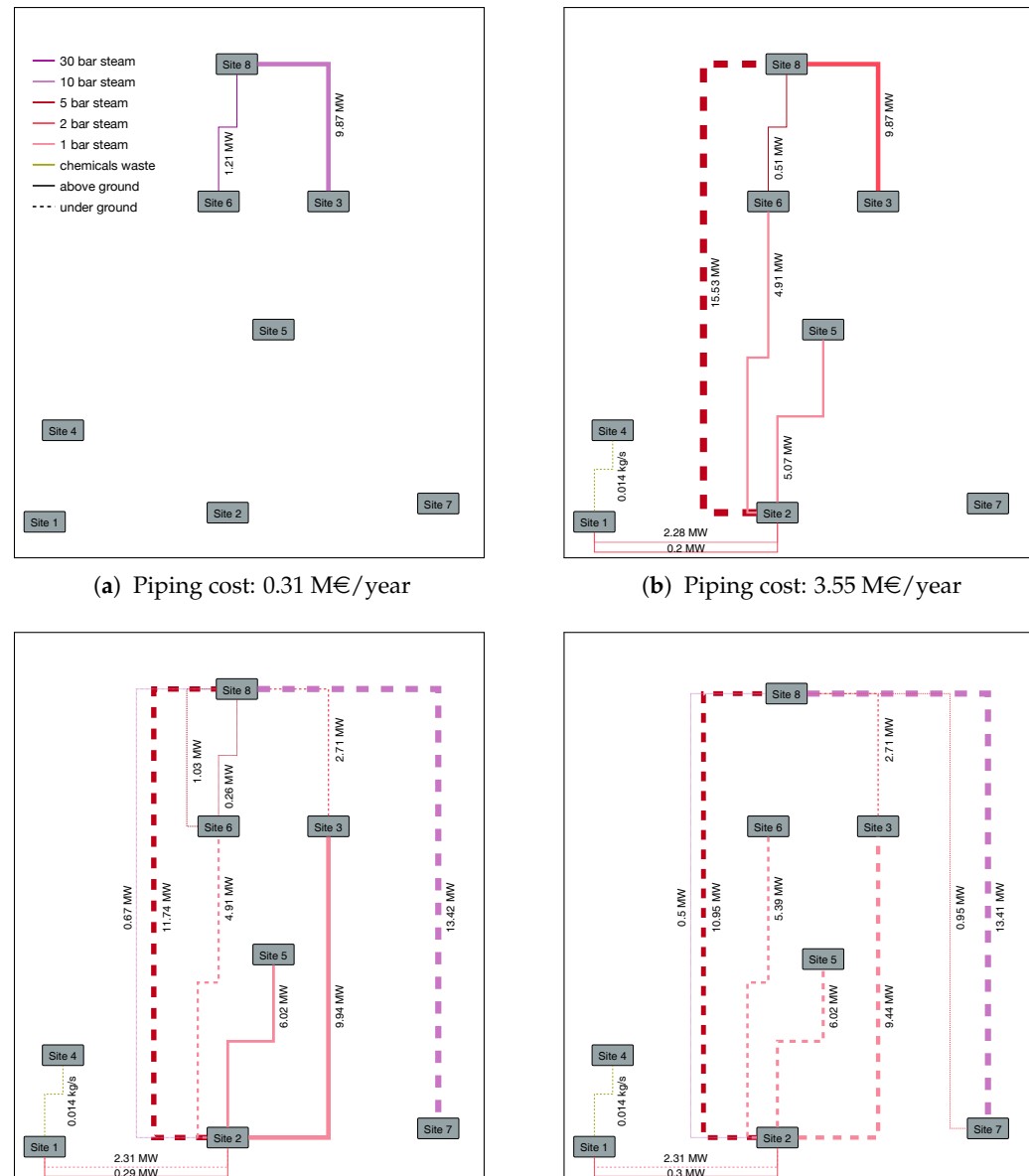

**Figure 8.** Optimal piping connections at different investment levels.

## 5.4. Comparison with Baseline and State of the Art

The baseline operation of the sites represent their current state in which little or no heat is recovered, better performing technologies (e.g., HPs) are not integrated and heat and resources are not shared between locations. The baseline fuel and electricity consumption of the sites is modelled according to [56,59]. State-of-the-art methods do not consider the location aspects, as discussed in Section 2; thus, processes in different locations can exchange heat and resources without any restriction. Table 11 compares economic and environmental key performance indicators (KPIs) of the overall system in the baseline state, when a state-of-the-art targeting method [44] is used and when the proposed method is used. The objective function is modified in this case to include the piping cost. Compared to the baseline, the total cost and $CO_2$ emissions of the system decrease by 21% and 35%, respectively, using the proposed method. This is partially due to the integration of new energy conversion technologies and partially because of heat and resource sharing between the locations as discussed in the previous sections. The reduction in the economic objective function and $CO_2$ emissions

reach 33% and 48%, respectively, using the targeting approach. Although the targeting method offers further reduction in the cost, it represents only the theoretical potential while the proposed method takes the practical constraints of locations and piping investment into account.

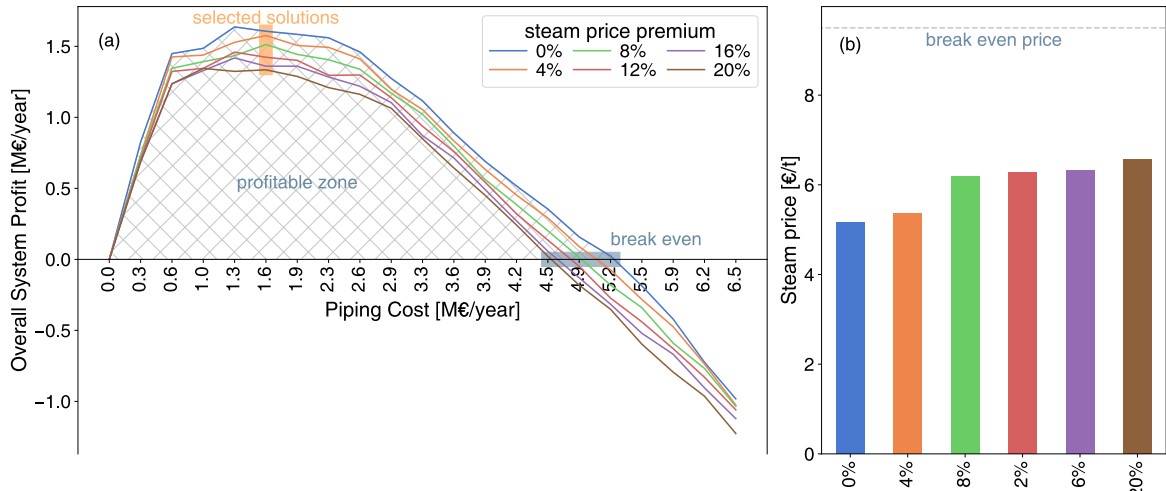

**Figure 9.** Overall profitability (**a**) and steam price (**b**) considering the premium charged by a third party.

**Table 11.** Comparison of the results of the present work with the baseline and targeting approach.

| KPI | Unit | Baseline | State of the Art | This Work |
|---|---|---|---|---|
| Operating cost | M€/year | 167.8 | 111.4 | 128.8 |
| Utility investment cost | M€/year | 0 | 1.1 | 1.3 |
| Piping cost | M€/year | 0 | 0 | 2.2 |
| Total | M€/year | 167.8 | 112.6 | 132.2 |
| $CO_2$ emissions | Mt/year | 867 | 452 | 562 |

## 6. Conclusions

This work proposes a PI method considering location aspects. Consequently, the heat cascade is reformulated to account for heat distribution losses and temperature drops, while the electricity balance is modified to include pumping work required to compensate pressure drops. The cost of the infrastructure between the plants is also considered in the form of piping cost and the resulting problem is formulated using MILP. Parametric optimisation is employed to systematically generate multiple solutions.

The method is first applied on a scenario with two chemical plants, to study the potential heat sharing between them. The results show that the lowest total cost solution is achieved by sharing 1 bar steam between the plants, with a cogeneration engine installed in one plant and a HP is integrated in each. Other solutions are also found, which prove the possibility of eliminating the main heating utility of one plant by multi-level steam sharing.

As a large-scale application of the method, parametric optimisation on eight industrial plants in geographical proximity is also completed. In this case, with small piping investment budgets, the optimal solutions favour sharing heat via high-pressure steam, since higher pressure levels require smaller pipe diameters. With larger budgets, lower pressure steam sharing options emerge, stemming from reduced heat losses. Following the same trend, above-ground pipes are preferred at low piping cost limits, while underground pipes are selected at high limits, due to the trade-off between heat losses and piping investment.

When process industries are not willing to take the risk of investing in inter-plant infrastructure, involvement of a third party can be beneficial. The third-party, making the initial investment and selling steam between plants, could be a non-profit governmental organisation or a utility company

with profitability targets. In either case, solutions resulting in overall system profit are obtained, with lower system profit at higher steam price premiums. Industries benefit from such a strategy by avoiding investment risks while benefitting from a 40% reduction in steam prices (on average) with the third party profiting from a steam price premium of up to 20%.

The optimal results obtained using the proposed method lead to a 21% and 35% reduction in the total cost and $CO_2$ emissions, respectively, compared to the baseline operation of the sites, resulting from heat and resource sharing between the sites and integrating new energy conversion systems. The theoretical optimum suggested by the targeting approach results in an even lower total cost and environmental impact; however, contrary to the work presented in this paper, it does not account for technical constraints (e.g., using commercially available technologies) or economic constraints (e.g., including the investment cost of the new technologies or the piping cost).

The present work provides a complete analysis for industrial symbiosis with heat and resource sharing as well as a set of options for investment budgets on inter-plant infrastructure. Heat losses and pumping work requirements are assumed to scale linearly with the flow for inclusion within the MILP framework. Such assumptions simplify the model solution process but may result in missing the global optimum solution. Thus, further analysis should be carried out, studying non-linearity aspects and their impact in the results. Moreover, large industrial retrofit projects, as the one presented in this work, are generally carried out over a long time horizon. Hence, future work should include investment scheduling analysis of the system, which will offer insight into the timeline of the investment in new technologies and piping as well as utility replacement requirements.

**Author Contributions:** Conceptualization, H.B., I.K. and F.M.; methodology, H.B.; software, H.B.; validation, H.B. and I.K.; formal analysis, H.B. and I.K.; investigation, H.B., I.K. and F.M.; resources, F.M.; data curation, H.B.; writing–original draft preparation, H.B.; writing–review and editing, H.B. and I.K.; visualization, H.B.; supervision, I.K. and F.M.; project administration, I.K. and F.M.; funding acquisition, F.M.

**Funding:** This research project is financially supported by the Swiss Innovation Agency lnnosuisse and is part of the Swiss Competence Center for Energy Research SCCER EIP. This project has received funding from the European Union's Horizon 2020 research and innovation programme under grant agreement No 679386. This work was supported by the Swiss State Secretariat for Education, Research and Innovation (SERI) under contract number 15.0217.

**Conflicts of Interest:** The authors declare no conflict of interest.

## Appendix A

*Appendix A.1 Heat Stream Data of the Processes*

The heat streams of the plants in the case study are listed in Tables A1–A7 respectively. The stream information includes inlet and outlet temperatures and inlet and outlet enthalpies.

**Table A1.** Site 1 heat streams.

| Stream | $T^{in}$ [°C] | $T^{out}$ [°C] | $H^{in}$ [kW] | $H^{out}$ [kW] |
|--------|------|------|------|------|
| s1p1 | 51 | 56 | 0 | 10 |
| s1p2 | 56 | 56 | 0 | 192 |
| s2p1 | 79 | 84 | 0 | 54 |
| s2p2 | 84 | 84 | 0 | 1032 |
| s3p1 | 61 | 66 | 0 | 7 |
| s3p2 | 66 | 66 | 0 | 135 |
| s4p1 | 58 | 63 | 0 | 47 |
| s4p2 | 63 | 63 | 0 | 900 |
| s5p1 | 60 | 65 | 0 | 49 |
| s5p2 | 65 | 65 | 0 | 931 |
| s6p1 | 43 | 100 | 0 | 49 |

**Table A1.** *Cont.*

| Stream | $T^{in}$ [°C] | $T^{out}$ [°C] | $H^{in}$ [kW] | $H^{out}$ [kW] |
|---|---|---|---|---|
| s6p2 | 100 | 100 | 0 | 938 |
| s7p1 | 43 | 100 | 0 | 21 |
| s7p2 | 100 | 100 | 0 | 391 |
| s8p1 | 63 | 68 | 0 | 36 |
| s8p2 | 68 | 68 | 0 | 685 |
| s9p1 | 53 | 58 | 0 | 11 |
| s9p2 | 58 | 58 | 0 | 200 |
| s10 | 53 | 31 | 305 | 0 |
| s11 | 50 | 38 | 23 | 0 |
| s12 | 41 | 32 | 1109 | 0 |
| s13 | 55 | 33 | 575 | 0 |
| s14 | 44 | 31 | 123 | 0 |
| s15 | 47 | 30 | 72 | 0 |
| s16p1 | 67 | 67 | 538 | 0 |
| s16p2 | 67 | 40 | 132 | 0 |
| s17p1 | 86 | 86 | 452 | 0 |
| s17p2 | 86 | 40 | 134 | 0 |
| s18p1 | 57 | 57 | 1084 | 0 |
| s18p2 | 57 | 35 | 160 | 0 |
| s19p1 | 67 | 67 | 526 | 0 |
| s19p2 | 67 | 40 | 117 | 0 |
| s20p1 | 67 | 67 | 319 | 0 |
| s20p2 | 67 | 40 | 71 | 0 |
| s21p1 | 69 | 69 | 881 | 0 |
| s21p2 | 69 | 40 | 247 | 0 |
| s22p1 | 177 | 61 | 354 | 0 |
| s22p2 | 61 | 61 | 840 | 0 |
| s22p3 | 61 | 40 | 93 | 0 |
| s23p1 | 81 | 81 | 290 | 0 |
| s23p2 | 81 | 40 | 75 | 0 |
| s24 | 58 | 40 | 340 | 0 |

**Table A2.** Site 2 heat streams.

| Stream | $T^{in}$ [°C] | $T^{out}$ [°C] | $H^{in}$ [kW] | $H^{out}$ [kW] |
|---|---|---|---|---|
| s1p1 | 135 | 140 | 0 | 95 |
| s1p2 | 140 | 140 | 0 | 1805 |
| s2p1 | 135 | 140 | 0 | 290 |
| s2p2 | 140 | 140 | 0 | 5510 |
| s3p1 | 150 | 155 | 0 | 640 |
| s3p2 | 155 | 155 | 0 | 12160 |
| s4p1 | 142 | 147 | 0 | 40 |
| s4p2 | 147 | 147 | 0 | 760 |
| s5p1 | 136 | 141 | 0 | 45 |
| s5p2 | 141 | 141 | 0 | 855 |
| s6p1 | 135 | 140 | 0 | 5 |
| s6p2 | 140 | 140 | 0 | 95 |

**Table A2.** *Cont.*

| Stream | $T^{in}$ [°C] | $T^{out}$ [°C] | $H^{in}$ [kW] | $H^{out}$ [kW] |
|---|---|---|---|---|
| s7p1 | 135 | 140 | 0 | 40 |
| s7p2 | 140 | 140 | 0 | 760 |
| s8p1 | 123 | 128 | 0 | 920 |
| s8p2 | 128 | 128 | 0 | 17480 |
| s9p1 | 114 | 119 | 0 | 85 |
| s9p2 | 119 | 119 | 0 | 1615 |
| s10p1 | 115 | 120 | 0 | 10 |
| s10p2 | 120 | 120 | 0 | 190 |
| s11p1 | 115 | 120 | 0 | 45 |
| s11p2 | 120 | 120 | 0 | 855 |
| s12p1 | 115 | 120 | 0 | 210 |
| s12p2 | 120 | 120 | 0 | 3990 |
| s13p1 | 115 | 120 | 0 | 15 |
| s13p2 | 120 | 120 | 0 | 285 |
| s28 | 49 | 44 | 200 | 0 |
| s29 | 50 | 45 | 100 | 0 |
| s30 | 51 | 46 | 100 | 0 |
| s31 | 53 | 48 | 1100 | 0 |
| s32 | 50 | 45 | 900 | 0 |
| s33 | 53 | 48 | 500 | 0 |
| s34 | 56 | 51 | 200 | 0 |
| s35 | 52 | 47 | 100 | 0 |
| s36 | 49 | 44 | 100 | 0 |
| s37 | 57 | 52 | 400 | 0 |
| s38 | 47 | 42 | 100 | 0 |
| s39 | 56 | 51 | 100 | 0 |
| s40 | 54 | 49 | 300 | 0 |
| s41 | 48 | 43 | 100 | 0 |
| s42p1 | 115 | 115 | 4950 | 0 |
| s42p2 | 115 | 110 | 550 | 0 |
| s43p1 | 112 | 112 | 19080 | 0 |
| s43p2 | 112 | 107 | 2120 | 0 |
| s44p1 | 115 | 115 | 360 | 0 |
| s44p2 | 115 | 110 | 40 | 0 |
| s45p1 | 111 | 111 | 1260 | 0 |
| s45p2 | 111 | 106 | 140 | 0 |
| s46p1 | 123 | 123 | 450 | 0 |
| s46p2 | 123 | 118 | 50 | 0 |
| s47p1 | 118 | 118 | 900 | 0 |
| s47p2 | 118 | 113 | 100 | 0 |
| s48p1 | 112 | 112 | 2205 | 0 |
| s48p2 | 112 | 107 | 245 | 0 |
| s49p1 | 113 | 113 | 1710 | 0 |
| s49p2 | 113 | 108 | 190 | 0 |
| s50p1 | 114 | 114 | 2970 | 0 |
| s50p2 | 114 | 109 | 330 | 0 |
| s51p1 | 116 | 116 | 3150 | 0 |

**Table A2.** *Cont.*

| Stream | T$^{in}$ [°C] | T$^{out}$ [°C] | H$^{in}$ [kW] | H$^{out}$ [kW] |
|--------|------|------|------|------|
| s51p2 | 116 | 111 | 350 | 0 |
| s52p1 | 127 | 127 | 720 | 0 |
| s52p2 | 127 | 122 | 80 | 0 |
| s53p1 | 109 | 109 | 720 | 0 |
| s53p2 | 109 | 104 | 80 | 0 |

**Table A3.** Site 3 heat streams.

| Stream | T$^{in}$ [°C] | T$^{out}$ [°C] | H$^{in}$ [kW] | H$^{out}$ [kW] |
|--------|------|------|------|------|
| s1 | 65 | 85 | 0 | 396 |
| s2 | 85 | 60 | 742 | 0 |
| s3 | 40 | 60 | 0 | 743 |
| s4 | 40 | 20 | 1121 | 0 |
| s5 | 15 | 85 | 0 | 63 |
| s6 | 15 | 60 | 0 | 46 |
| s7 | 15 | 85 | 0 | 2022 |
| s8 | 35 | 60 | 0 | 627 |
| s9 | 1 | 1 | 672 | 0 |
| s10 | 6 | 1 | 1122 | 0 |
| s11 | 1 | 15 | 0 | 1291 |
| s12 | 70 | 5 | 7631 | 0 |
| s13 | 15 | 80 | 0 | 7624 |
| s14 | 48 | 65 | 0 | 296 |
| s15 | 48 | 65 | 0 | 660 |
| s16 | 15 | 52 | 0 | 581 |
| s17 | 15 | 55 | 0 | 1294 |
| s18 | 65 | 75 | 0 | 579 |
| s19 | 105 | 100 | 27 | 0 |
| s20 | 100 | 100 | 5478 | 0 |
| s21 | 100 | 25 | 758 | 0 |
| s22 | 15 | 80 | 0 | 5419 |
| s23 | 78 | 105 | 0 | 2612 |
| s24 | 105 | 105 | 0 | 5214 |
| s25 | 5 | 80 | 0 | 1347 |
| s26 | 103 | 10 | 8536 | 0 |
| s27 | 10 | 10 | 840 | 0 |
| s28 | 10 | 6 | 367 | 0 |
| s29 | 6 | 6 | 668 | 0 |

**Table A4.** Site 4 heat streams.

| Stream | T$^{in}$ [°C] | T$^{out}$ [°C] | H$^{in}$ [kW] | H$^{out}$ [kW] |
|--------|------|------|------|------|
| s1 | 1450 | 650 | 37 | 0 |
| s2 | 650 | 250 | 37 | 0 |
| s3 | 250 | 100 | 37 | 0 |
| s4 | 860 | 850 | 348 | 0 |
| s5 | 890 | 880 | 668 | 0 |

**Table A4.** *Cont.*

| Stream | $T^{in}$ [°C] | $T^{out}$ [°C] | $H^{in}$ [kW] | $H^{out}$ [kW] |
|--------|------|------|------|------|
| s6 | 1531 | 1504 | 268 | 0 |
| s7 | 860 | 700 | 1268 | 0 |
| s8 | 1719 | 1531 | 328 | 0 |
| s9 | 880 | 860 | 133 | 0 |
| s10 | 560 | 390 | 1468 | 0 |
| s11 | 104 | 50 | 2115 | 0 |
| s12 | 250 | 100 | 37 | 0 |
| s13 | 1822 | 1719 | 387 | 0 |
| s14 | 1222 | 1177 | 89 | 0 |
| s15 | 1177 | 1050 | 30 | 0 |
| s16 | 390 | 104 | 1597 | 0 |
| s17 | 104 | 45 | 9 | 0 |
| s18 | 650 | 250 | 37 | 0 |
| s19 | 700 | 560 | 1166 | 0 |
| s20 | 1000 | 1000 | 37 | 0 |
| s21 | 2000 | 2000 | 566 | 0 |
| s22 | 1504 | 1222 | 357 | 0 |
| s23 | 1450 | 650 | 37 | 0 |
| s24 | 2000 | 1822 | 953 | 0 |
| s25 | 25 | 104 | 0 | 113 |
| s26 | 25 | 56 | 0 | 1255 |
| s27 | 85 | 100 | 0 | 1255 |
| s28 | 390 | 104 | 2652 | 0 |
| s29 | 100 | 100 | 0 | 1255 |
| s30 | 56 | 85 | 0 | 1255 |
| s31 | 390 | 104 | 2477 | 0 |
| s32 | 100 | 104 | 0 | 1255 |
| s33 | 560 | 390 | 9018 | 0 |
| s34 | 700 | 560 | 7807 | 0 |
| s35 | 880 | 860 | 1200 | 0 |
| s36 | 860 | 700 | 9297 | 0 |
| s37 | 1177 | 1050 | 2710 | 0 |
| s38 | 2000 | 1822 | 3124 | 0 |
| s39 | 1719 | 1531 | 3882 | 0 |
| s40 | 1504 | 1222 | 5864 | 0 |
| s41 | 1531 | 1504 | 338 | 0 |
| s42 | 1222 | 1177 | 895 | 0 |
| s43 | 2000 | 2000 | 606 | 0 |
| s44 | 1822 | 1719 | 1954 | 0 |
| s45 | 25 | 40 | 0 | 62 |
| s46 | 25 | 60 | 0 | 26 |
| s47 | 626 | 850 | 0 | 11514 |
| s48 | 850 | 850 | 0 | 39975 |
| s49 | 850 | 860 | 0 | 348 |
| s50 | 850 | 890 | 0 | 541 |
| s51 | 1100 | 1100 | 45208 | 0 |
| s52 | 1100 | 890 | 11358 | 0 |

**Table A4.** *Cont.*

| Stream | T$^{in}$ [°C] | T$^{out}$ [°C] | H$^{in}$ [kW] | H$^{out}$ [kW] |
|---|---|---|---|---|
| s53 | 25 | 400 | 0 | 37 |
| s54 | 25 | 1000 | 0 | 37 |
| s55 | 25 | 225 | 0 | 37 |
| s56 | 150 | 35 | 3613 | 0 |
| s57 | 225 | 35 | 1598 | 0 |
| s58 | 104 | 35 | 1300 | 0 |
| s59 | 115 | 35 | 1287 | 0 |
| s60 | 400 | 35 | 3126 | 0 |
| s61 | 390 | 150 | 8202 | 0 |
| s62 | 50 | 268 | 0 | 9018 |
| s63 | 268 | 437 | 0 | 7807 |
| s64 | 437 | 626 | 0 | 9297 |
| s65 | 850 | 850 | 0 | 1200 |
| s66 | 850 | 850 | 1 | 0 |
| s67 | 910 | 910 | 0 | 2401 |
| s68 | 910 | 1027 | 0 | 3609 |
| s69 | 850 | 900 | 0 | 1651 |
| s70 | 1377 | 1450 | 0 | 2281 |
| s71 | 1027 | 910 | 146 | 0 |
| s72 | 1227 | 1227 | 0 | 921 |
| s73 | 910 | 900 | 25 | 0 |
| s74 | 900 | 850 | 153 | 0 |
| s75 | 1327 | 1377 | 0 | 1509 |
| s76 | 900 | 910 | 0 | 324 |
| s77 | 900 | 900 | 0 | 596 |
| s78 | 1377 | 1377 | 0 | 843 |
| s79 | 1227 | 1027 | 85 | 0 |
| s80 | 850 | 850 | 0 | 1212 |
| s81 | 1027 | 1027 | 5525 | 0 |
| s82 | 1327 | 1327 | 0 | 445 |
| s83 | 1450 | 1450 | 0 | 606 |
| s84 | 1027 | 1227 | 0 | 5948 |
| s85 | 1227 | 1327 | 0 | 2961 |

**Table A5.** Site 5 heat streams.

| Stream | T$^{in}$ [°C] | T$^{out}$ [°C] | H$^{in}$ [kW] | H$^{out}$ [kW] |
|---|---|---|---|---|
| s1 | 66 | 98 | 0 | 12 |
| s2 | 98 | 4 | 35 | 0 |
| s3 | 86 | 4 | 277 | 0 |
| s4 | 4 | 66 | 0 | 236 |
| s5 | 66 | 86 | 0 | 68 |
| s6 | 6 | 4 | 8 | 0 |
| s7 | 69 | 69 | 0 | −1051 |
| s8 | 61 | 61 | 0 | 988 |
| s9 | 66 | 15 | 94 | 0 |
| s10 | 4 | 66 | 0 | 236 |

**Table A5.** *Cont.*

| Stream | T$^{in}$ [°C] | T$^{out}$ [°C] | H$^{in}$ [kW] | H$^{out}$ [kW] |
|--------|------|------|------|------|
| s11 | 70 | 70 | 0 | 1051 |
| s12 | 66 | 66 | 0 | −1005 |
| s13 | 60 | 60 | 0 | −988 |
| s14 | 66 | 66 | 0 | 1005 |
| s15 | 60 | 15 | 81 | 0 |
| s16 | 69 | 15 | 102 | 0 |
| s17 | 98 | 4 | 35 | 0 |
| s18 | 20 | 10 | 73 | 0 |
| s19 | 6 | 4 | 8 | 0 |
| s20 | 4 | 20 | 0 | 118 |
| s21 | −21 | −25 | 11 | 0 |
| s22 | 7 | −21 | 67 | 0 |
| s23 | −21 | −21 | 0 | −60 |
| s24 | 4 | 95 | 0 | 342 |
| s25 | 6 | 4 | 8 | 0 |
| s26 | 83 | 3 | 254 | 0 |
| s27 | 20 | 73 | 0 | 166 |
| s28 | −6 | −6 | 1 | 0 |
| s29 | 73 | 83 | 0 | 32 |
| s30 | 3 | −6 | 26 | 0 |
| s31 | −6 | −6 | 0 | −197 |
| s32 | −6 | −35 | 80 | 0 |
| s33 | 75 | 15 | 13 | 0 |
| s34 | 5 | 5 | 30 | 0 |
| s35 | 15 | 55 | 0 | 17 |
| s36 | 67 | 80 | 0 | 21 |
| s37 | 65 | 15 | 10 | 0 |
| s38 | 59 | 70 | 0 | 19 |
| s39 | 69 | 15 | 102 | 0 |
| s40 | 4 | 90 | 0 | 327 |
| s41 | 66 | 15 | 94 | 0 |
| s42 | 100 | 170 | 0 | 447 |
| s43 | 100 | 170 | 0 | 165 |
| s44 | 170 | 170 | 0 | 3057 |
| s45 | 35 | 35 | 0 | 122 |
| s46 | 0 | −3 | 205 | 0 |
| s47 | 5 | 1 | 1105 | 0 |
| s48 | 69 | 75 | 0 | 116 |
| s49 | 105 | 105 | 0 | 472 |
| s50 | 170 | 170 | 0 | 1132 |
| s51 | 44 | 25 | 713 | 0 |
| s52 | 9 | 26 | 0 | 300 |
| s53 | 170 | 190 | 0 | 73 |
| s54 | 78 | 78 | 0 | 179 |
| s55 | 44 | 44 | 0 | −2260 |
| s56 | 35 | 35 | 0 | 551 |
| s57 | 79 | 85 | 0 | 18 |

**Table A5.** *Cont.*

| Stream | $T^{in}$ [°C] | $T^{out}$ [°C] | $H^{in}$ [kW] | $H^{out}$ [kW] |
|--------|------|------|------|------|
| s58 | 6 | 28 | 0 | 1110 |
| s59 | 95 | 95 | 0 | 179 |
| s60 | 48 | 75 | 0 | 1238 |
| s61 | 75 | 4 | 3257 | 0 |
| s62 | 6 | 48 | 0 | 2046 |
| s63 | 66 | 76 | 0 | 195 |
| s64 | 85 | 85 | 0 | 857 |
| s65 | 54 | 4 | 151 | 0 |
| s66 | 44 | 5 | 118 | 0 |
| s67 | 170 | 190 | 0 | 27 |
| s68 | 70 | 70 | 0 | 225 |
| s69 | 15 | 55 | 0 | 145 |
| s70 | 74 | 80 | 0 | 303 |
| s71 | 32 | 25 | 632 | 0 |
| s72 | 86 | 4 | 277 | 0 |
| s73 | 66 | 86 | 0 | 68 |

**Table A6.** Site 6 heat streams.

| Stream | $T^{in}$ [°C] | $T^{out}$ [°C] | $H^{in}$ [kW] | $H^{out}$ [kW] |
|--------|------|------|------|------|
| s1 | 5 | 35 | 0 | 36.88 |
| s2 | 15 | 35 | 0 | 1064.17 |
| s3 | 36 | 35 | 0 | −305.55 |
| s4 | 36 | 100 | 0 | 273.15 |
| s5 | 100 | 100 | 0 | 2297.7 |
| s6 | 36 | 90 | 0 | 115.9 |
| s7 | 148 | 25 | 0 | −883.18 |
| s8 | 115 | 115 | 0 | −391.55 |
| s9 | 148 | 148 | 0 | −3622.26 |
| s10 | 115 | 25 | 0 | −66.94 |
| s11 | 100 | 25 | 0 | −106.43 |
| s12 | 100 | 100 | 0 | −767.76 |
| s13 | 50 | 138 | 0 | 270.27 |
| s14 | 138 | 144 | 0 | 9.35 |
| s15 | 138 | 138 | 0 | 1560.4 |
| s16 | 77 | 78 | 0 | 391.25 |
| s17 | 5 | 89 | 0 | 599.1 |
| s18 | 5 | 78 | 0 | 67.25 |
| s19 | 78 | 115 | 0 | 574.64 |
| s20 | 5 | 48 | 0 | 1571.1 |
| s21 | 5 | 78 | 0 | 134.49 |
| s22 | 51 | 48 | 0 | −397.36 |
| s23 | 51 | 89 | 0 | 140.6 |
| s24 | 5 | 48 | 0 | 85.59 |
| s25 | 40 | 40 | 0 | 76.23 |
| s26 | 5 | 138 | 0 | 38.88 |
| s27 | 35 | 40 | 0 | 114.35 |

**Table A6.** *Cont.*

| Stream | $T^{in}$ [°C] | $T^{out}$ [°C] | $H^{in}$ [kW] | $H^{out}$ [kW] |
|---|---|---|---|---|
| s28 | 5 | 40 | 0 | 11.43 |
| s29 | 138 | 138 | 0 | 151.7 |
| s30 | 70 | 99 | 0 | 0.59 |
| s31 | 52 | 20 | 0 | −0.42 |
| s32 | 51 | 20 | 0 | −0.07 |
| s33 | 20 | 51 | 0 | 0.07 |
| s34 | 20 | 99 | 0 | 0.77 |
| s35 | 99 | 130 | 0 | 2.45 |
| s36 | 51 | 59 | 0 | 0.15 |
| s37 | 49 | 20 | 0 | −0.55 |
| s38 | 51 | 59 | 0 | 0.15 |
| s39 | 49 | 20 | 0 | −0.55 |
| s40 | 20 | 100 | 0 | 0.13 |
| s41 | 20 | 49 | 0 | 0.55 |
| s42 | 20 | 50 | 0 | 0.04 |
| s43 | 20 | 50 | 0 | 0.04 |
| s44 | 20 | 59 | 0 | 0.01 |
| s45 | 20 | 59 | 0 | 0.01 |
| s46 | 51 | 20 | 0 | −0.07 |
| s47 | 50 | 20 | 0 | −0.04 |
| s48 | 50 | 20 | 0 | −0.04 |
| s49 | 20 | 49 | 0 | 0.55 |
| s50 | 20 | 51 | 0 | 0.07 |
| s51 | 85 | 85 | 2.27 | 0 |
| s52 | 88 | 88 | 0.57 | 0 |
| s53 | 88 | 20 | 0.05 | 0 |
| s54 | 84 | 20 | 0.05 | 0 |
| s55 | 54 | 54 | 0 | 1.77 |
| s56 | 54 | 54 | 0 | 0.46 |
| s57 | 108 | 108 | 0 | 0.49 |
| s58 | 103 | 103 | 0 | 2.3 |
| s59 | 99 | 20 | 0.34 | 0 |
| s60 | 71 | 20 | 0.05 | 0 |
| s61 | 69 | 20 | 0.04 | 0 |
| s62 | 79 | 20 | 0.08 | 0 |
| s63 | 54 | 20 | 0.11 | 0 |
| s64 | 54 | 20 | 0.03 | 0 |
| s65 | 85 | 85 | 0 | 2.34 |
| s66 | 88 | 88 | 0 | 0.54 |
| s67 | 70 | 70 | 1.76 | 0 |
| s68 | 70 | 20 | 0.16 | 0 |
| s69 | 71 | 71 | 0 | 0.75 |
| s70 | 70 | 70 | 0 | 2.03 |
| s71 | 62 | 62 | 0.66 | 0 |
| s72 | 84 | 84 | 0 | 0.45 |
| s73 | 88 | 88 | 0.39 | 0 |
| s74 | 62 | 62 | 0 | 0.54 |

**Table A6.** *Cont.*

| Stream | T$^{in}$ [°C] | T$^{out}$ [°C] | H$^{in}$ [kW] | H$^{out}$ [kW] |
| --- | --- | --- | --- | --- |
| s75 | 69 | 69 | 0.47 | 0 |
| s76 | 71 | 71 | 0.59 | 0 |
| s77 | 99 | 99 | 2.3 | 0 |
| s78 | 88 | 20 | 0.07 | 0 |
| s79 | 84 | 84 | 0.44 | 0 |
| s80 | 69 | 69 | 0 | 0.5 |
| s81 | 54 | 54 | 1.81 | 0 |
| s82 | 88 | 88 | 0 | 0.54 |
| s83 | 79 | 79 | 0 | 0.91 |
| s84 | 62 | 20 | 0.05 | 0 |
| s85 | 108 | 108 | 0.48 | 0 |
| s86 | 79 | 79 | 0.79 | 0 |
| s87 | 108 | 20 | 0.08 | 0 |
| s88 | 54 | 54 | 0.43 | 0 |
| s89 | 85 | 20 | 0.27 | 0 |
| s90 | 100 | 100 | 1.72 | 0 |
| s91 | 58 | 90 | 0 | 0.11 |
| s92 | 100 | 100 | 0 | 1.61 |
| s93 | 20 | 61 | 0 | 0.02 |
| s94 | 20 | 69 | 0 | 0.02 |
| s95 | 47 | 61 | 0 | 0.31 |
| s96 | 53 | 69 | 0 | 0.26 |
| s97 | 20 | 160 | 0 | 0.29 |

**Table A7.** Site 7 heat streams.

| Stream | T$^{in}$ [°C] | T$^{out}$ [°C] | H$^{in}$ [kW] | H$^{out}$ [kW] |
| --- | --- | --- | --- | --- |
| s1 | 25 | 110 | 0 | 72.53 |
| s2 | 145 | 145 | 0 | 359.51 |
| s3 | 180 | 180 | 0 | 28.07 |
| s4 | 160 | 180 | 0 | 141.91 |
| s5 | 110 | 150 | 0 | 97.76 |
| s6 | 70 | 70 | 0 | 116.68 |
| s7 | 180 | 190 | 0 | 141.91 |
| s8 | 66 | 64 | 3.15 | 0 |
| s9 | 141 | 141 | 0 | −6.31 |
| s10 | 58 | 55 | 3.15 | 0 |
| s11 | 86 | 75 | 15.77 | 0 |
| s12 | 132 | 123 | 15.77 | 0 |
| s13 | 65 | 65 | 0 | 34.69 |
| s14 | 189 | 160 | 12.61 | 0 |
| s15 | 25 | 26 | 0 | 9.46 |
| s16 | 139 | 133 | 12.61 | 0 |
| s17 | 108 | 99 | 15.77 | 0 |
| s18 | 62 | 59 | 3.15 | 0 |
| s19 | 99 | 95 | 15.77 | 0 |
| s20 | 93 | 77 | 12.61 | 0 |

**Table A7.** *Cont.*

| Stream | $T^{in}$ [°C] | $T^{out}$ [°C] | $H^{in}$ [kW] | $H^{out}$ [kW] |
|--------|------|------|-------|-------|
| s21 | 139 | 138 | 6.31 | 0 |
| s22 | 178 | 175 | 9.46 | 0 |
| s23 | 153 | 132 | 15.77 | 0 |
| s24 | 118 | 114 | 15.77 | 0 |
| s25 | 59 | 58 | 3.15 | 0 |
| s26 | 181 | 178 | 9.46 | 0 |
| s27 | 146 | 139 | 12.61 | 0 |
| s28 | 63 | 63 | 0 | 25.23 |
| s29 | 69 | 66 | 3.15 | 0 |
| s30 | 112 | 106 | 12.61 | 0 |
| s31 | 152 | 130 | 15.77 | 0 |
| s32 | 145 | 143 | 6.31 | 0 |
| s33 | 120 | 114 | 15.77 | 0 |
| s34 | 180 | 180 | 0 | 44.15 |
| s35 | 140 | 139 | 6.31 | 0 |
| s36 | 140 | 140 | 0 | $-6.31$ |
| s37 | 125 | 112 | 12.61 | 0 |
| s38 | 175 | 167 | 9.46 | 0 |
| s39 | 78 | 54 | 12.61 | 0 |
| s40 | 55 | 52 | 3.15 | 0 |
| s41 | 148 | 138 | 12.61 | 0 |
| s42 | 75 | 53 | 15.77 | 0 |
| s43 | 128 | 124 | 12.61 | 0 |
| s44 | 113 | 104 | 15.77 | 0 |
| s45 | 110 | 110 | 0 | 15.77 |
| s46 | 95 | 86 | 15.77 | 0 |
| s47 | 160 | 146 | 12.61 | 0 |
| s48 | 25 | 26 | 0 | 18.92 |
| s49 | 178 | 152 | 15.77 | 0 |
| s50 | 52 | 45 | 3.15 | 0 |
| s51 | 77 | 46 | 12.61 | 0 |
| s52 | 134 | 128 | 12.61 | 0 |
| s53 | 84 | 61 | 15.77 | 0 |
| s54 | 114 | 108 | 15.77 | 0 |
| s55 | 123 | 118 | 15.77 | 0 |
| s56 | 119 | 114 | 12.61 | 0 |
| s57 | 63 | 110 | 0 | 25.23 |
| s58 | 114 | 104 | 12.61 | 0 |
| s59 | 142 | 141 | 6.31 | 0 |
| s60 | 143 | 142 | 6.31 | 0 |
| s61 | 138 | 134 | 12.61 | 0 |
| s62 | 130 | 120 | 15.77 | 0 |
| s63 | 104 | 100 | 15.77 | 0 |
| s64 | 100 | 93 | 15.77 | 0 |
| s65 | 93 | 84 | 15.77 | 0 |
| s66 | 110 | 112 | 0 | 47.3 |
| s67 | 64 | 63 | 3.15 | 0 |

**Table A7.** *Cont.*

| Stream | $T^{in}$ [°C] | $T^{out}$ [°C] | $H^{in}$ [kW] | $H^{out}$ [kW] |
|--------|-----|-----|--------|---------|
| s68 | 114 | 113 | 15.77 | 0 |
| s69 | 133 | 125 | 12.61 | 0 |
| s70 | 104 | 94 | 12.61 | 0 |
| s71 | 106 | 93 | 12.61 | 0 |
| s72 | 141 | 140 | 6.31 | 0 |
| s73 | 141 | 141 | 0 | −6.31 |
| s74 | 75 | 69 | 3.15 | 0 |
| s75 | 124 | 119 | 12.61 | 0 |
| s76 | 94 | 78 | 12.61 | 0 |
| s77 | 63 | 62 | 3.15 | 0 |
| s78 | 141 | 141 | 0 | −6.31 |
| s79 | 25 | 26 | 0 | 132.45 |
| s80 | 25 | 26 | 0 | 91.45 |
| s81 | 550 | 500 | 100.91 | 0 |
| s82 | 350 | 340 | 31.54 | 0 |
| s83 | 75 | 75 | 0 | 56.76 |
| s84 | 80 | 80 | 0 | 22.08 |
| s85 | 80 | 115 | 0 | 22.08 |
| s86 | 150 | 160 | 0 | 72.53 |
| s87 | 170 | 180 | 0 | 72.53 |
| s88 | 190 | 200 | 0 | 37.84 |
| s89 | 160 | 170 | 0 | 72.53 |
| s90 | 120 | 29 | 126.14 | 0 |
| s91 | 305 | 300 | 9.46 | 0 |
| s92 | 320 | 310 | 50.46 | 0 |
| s93 | 25 | 26 | 0 | 15.77 |
| s94 | 79 | 50 | 97.76 | 0 |
| s95 | 75 | 80 | 0 | 12.61 |
| s96 | 120 | 120 | 0 | 41 |
| s97 | 25 | 26 | 0 | 22.08 |
| s98 | 340 | 330 | 15.77 | 0 |
| s99 | 550 | 550 | 0 | −100.91 |
| s100 | 157 | 120 | 167.14 | 0 |
| s101 | 214 | 140 | 135.6 | 0 |
| s102 | 50 | 41 | 63.07 | 0 |
| s103 | 168 | 120 | 69.38 | 0 |
| s104 | 180 | 190 | 0 | 110.38 |
| s105 | 140 | 94 | 75.69 | 0 |
| s106 | 120 | 33 | 25.23 | 0 |
| s107 | 330 | 320 | 25.23 | 0 |
| s108 | 310 | 305 | 31.54 | 0 |
| s109 | 150 | 155 | 0 | 25.23 |
| s110 | 160 | 185 | 0 | 18.92 |
| s111 | 120 | 125 | 0 | 25.23 |
| s112 | 166 | 170 | 0 | 37.84 |
| s113 | 185 | 190 | 0 | 28.38 |
| s114 | 135 | 145 | 0 | 28.38 |

**Table A7.** *Cont.*

| Stream | T$^{in}$ [°C] | T$^{out}$ [°C] | H$^{in}$ [kW] | H$^{out}$ [kW] |
|--------|------|------|------|------|
| s115 | 110 | 110 | 0 | 56.76 |
| s116 | 97 | 100 | 0 | 9.46 |
| s117 | 93 | 95 | 0 | 53.61 |
| s118 | 185 | 185 | 0 | 47.3 |
| s119 | 125 | 130 | 0 | 47.3 |
| s120 | 155 | 166 | 0 | 12.61 |
| s121 | 130 | 135 | 0 | 37.84 |
| s122 | 25 | 26 | 0 | 22.08 |
| s123 | 90 | 92 | 0 | 18.92 |
| s124 | 145 | 150 | 0 | 31.54 |
| s125 | 110 | 45 | 47.3 | 0 |
| s126 | 95 | 97 | 0 | 63.07 |
| s127 | 92 | 93 | 0 | 34.69 |
| s128 | 65 | 35 | 227.06 | 0 |
| s129 | 25 | 26 | 0 | 15.77 |
| s130 | 114 | 45 | 72.53 | 0 |
| s131 | 180 | 190 | 0 | 28.38 |
| s132 | 160 | 180 | 0 | 28.38 |
| s133 | 60 | 60 | 0 | 217.6 |
| s134 | 130 | 130 | 0 | 25.23 |
| s135 | 25 | 26 | 0 | 22.08 |
| s136 | 118 | 129 | 0 | 31.54 |
| s137 | 185 | 190 | 0 | 34.69 |
| s138 | 129 | 134 | 0 | 12.61 |
| s139 | 25 | 26 | 0 | 15.77 |
| s140 | 93 | 29 | 47.3 | 0 |
| s141 | 73 | 44 | 230.21 | 0 |
| s142 | 160 | 185 | 0 | 34.69 |
| s143 | 295 | 295 | 0 | −3.15 |
| s144 | 116 | 118 | 0 | 18.92 |
| s145 | 104 | 48 | 85.15 | 0 |
| s146 | 185 | 185 | 0 | 72.53 |
| s147 | 218 | 217 | 56.76 | 0 |
| s148 | 325 | 315 | 6.31 | 0 |
| s149 | 99 | 35 | 28.38 | 0 |
| s150 | 315 | 315 | 0 | −31.54 |
| s151 | 180 | 180 | 0 | 56.76 |
| s152 | 165 | 170 | 0 | 28.38 |
| s153 | 295 | 290 | 56.76 | 0 |
| s154 | 204 | 203 | 91.45 | 0 |
| s155 | 25 | 26 | 0 | 6.31 |
| s156 | 80 | 85 | 0 | 12.61 |
| s157 | 79 | 45 | 258.59 | 0 |
| s158 | 615 | 615 | 0 | −280.67 |
| s159 | 50 | 29 | 25.23 | 0 |
| s160 | 135 | 145 | 0 | 56.76 |
| s161 | 312 | 300 | 37.84 | 0 |

**Table A7.** *Cont.*

| Stream | $T^{in}$ [°C] | $T^{out}$ [°C] | $H^{in}$ [kW] | $H^{out}$ [kW] |
|---|---|---|---|---|
| s162 | 203 | 197 | 22.08 | 0 |
| s163 | 49 | 49 | 0 | 34.69 |
| s164 | 125 | 125 | 0 | 189.21 |
| s165 | 171 | 130 | 37.84 | 0 |
| s166 | 138 | 60 | 88.3 | 0 |
| s167 | 130 | 135 | 0 | 28.38 |
| s168 | 130 | 45 | 69.38 | 0 |
| s169 | 150 | 160 | 0 | 28.38 |
| s170 | 125 | 130 | 0 | 28.38 |
| s171 | 170 | 180 | 0 | 28.38 |
| s172 | 25 | 26 | 0 | 9.46 |
| s173 | 145 | 150 | 0 | 44.15 |
| s174 | 300 | 295 | 56.76 | 0 |
| s175 | 180 | 180 | 0 | −31.54 |
| s176 | 160 | 165 | 0 | 44.15 |
| s177 | 315 | 314 | 18.92 | 0 |
| s178 | 60 | 70 | 0 | 12.61 |
| s179 | 70 | 80 | 0 | 15.77 |
| s180 | 127 | 127 | 0 | 15.77 |
| s181 | 75 | 76 | 0 | 18.92 |
| s182 | 314 | 312 | 18.92 | 0 |
| s183 | 59 | 45 | 56.76 | 0 |
| s184 | 160 | 160 | 0 | 15.77 |
| s185 | 115 | 125 | 0 | 22.08 |
| s186 | 170 | 180 | 0 | 47.3 |
| s187 | 115 | 115 | 0 | 88.3 |
| s188 | 180 | 190 | 0 | 18.92 |
| s189 | 78 | 45 | 113.53 | 0 |
| s190 | 25 | 26 | 0 | 9.46 |
| s191 | 170 | 170 | 0 | 28.38 |
| s192 | 85 | 85 | 0 | 15.77 |
| s193 | 25 | 26 | 0 | 15.77 |
| s194 | 70 | 45 | 59.92 | 0 |
| s195 | 94 | 34 | 85.15 | 0 |
| s196 | 120 | 29 | 81.99 | 0 |
| s197 | 180 | 60 | 5695.03 | 0 |
| s198 | 60 | 25 | 1601.73 | 0 |
| s199 | 900 | 340 | 25627.65 | 0 |

## Appendix A.2 Cement Fuel Properties

Table A8 depicts the LHV, share in the fuel mix and price of the fuels used in cement production.

**Table A8.** Cement fuel properties, shares and parices.

| Fuel | LHV [kJ/kg] | Share [%] | Price [€/kg] |
|---|---|---|---|
| Coal | 28066 | 10.5 | 60 |
| Lignite | 9500 | 23.8 | 11 |

**Table A8.** *Cont.*

| Fuel | LHV [kJ/kg] | Share [%] | Price [€/kg] |
|---|---|---|---|
| Petcoke | 32701 | 3.4 | 83 |
| Fuel oil | 40800 | 0.3 | 413 |
| Tyres | 30000 | 9.2 | 10 |
| Waste oil | 35000 | 2.5 | 10 |
| Waste paper | 17000 | 1.1 | 20 |
| Waste plastic | 20000 | 10.1 | 20 |
| Waste textile | 35000 | 0.4 | 20 |
| Other waste | 6000 | 18.7 | 20 |
| Animal meal | 20000 | 4 | 5 |
| Sewage | 15000 | 12.3 | 10 |
| Scrap wood | 17000 | 0.1 | 20 |
| Solvents | 25000 | 2.8 | 20 |

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
