# Peer review of "Incorporating Location Aspects in Process Integration Methodology"

_energies, doi:10.3390/en12173338_

Round 1
Reviewer 1 Report
The manuscript presents a Process Integration (PI) model considering the plant location using MILP technique. The objective function is the total cost; however, the piping cost is excluded. A cluster of several plants has been studied as a case study to examine the proposed model. The idea of considering location in PI studies is interesting itself while the manuscript requires Major Revision to be accepted for publication in Energies.
1- The literature survey needs to be improved. Are the authors aware of recent publications id Total Site targeting and Total Site optimization? Or the publications presented piping cost in heat exchanger networks. The gap of the literature must clearly be discussed.
2- The aim and the novelty of the manuscript must be highlighted.
3- I alluded to the difficult for MILP methods to reach the Global Minimum in constrained non-linear problems. The reviewer would have like to have seen some discussion of this.
4- In the case study, eight Total Site plants are considered in a cluster. However, as the processes are diverse with different level of energy demands, the diversity of utilities is not enough which will affect the results.
5- The reviewer would like to see how the proposed model may improve emissions reduction.
6- A few references are missing within the text.
Reviewer 2 Report
A well written paper that tackles a difficult but important industrial problem of including location in process integration design. The authors clearly present the method, despite it's complexity and include a case study. Some of the figures may be hard to read due to the small text of the annotations, however in any final manuscript this might not be a problem.
Reviewer 3 Report
I would like to summarise the strength of the reviewed work as:
A novel MILP formulation considering location aspect
Including heat losses, temperature and pressure drop and piping cost in process integration
Optimisation of intra and inter plant energy and resource recovery
Systematic generation of multiple solutions offering different retrofit options
20% improvement in the energy bill of an industrial cluster
As it was highlighted in the end of the conclusion section, the assumpions about heat losses and pumping requirement for inter-plant exchanges scale linearly with the flows to be able to keep the formulation as MILP. An impact of this assumption could be studied in a future work.
I recommend ths work for publication in the presented form.
Reviewer 4 Report
The submitted paper manuscript presents a very thorough and exhaustive process integration study taking into account the specific location of various plants on the energy balancing and investment cost of the energy equipment. The concept of the paper manuscript according to my knowledge is quite innovative and this is proved by the detailed "state-of-the-art" included in the paper manuscript. The entire paper manuscript is well-organized into appropriate sections and both the mathematical model and the corresponding results are well-analyzed. According to my humble opinion the technical integrity and the organization of the paper manuscript are above average. I have only a minor editorial comment: Authors are kindly advised to download "Energies" paper template and editorial instructions and to modify their paper manuscript accordingly to conform with journal's specific editorial requirements.
Round 2
Reviewer 1 Report
The reviewer's comments are well addressed.
I would recommend the authors to perform a final English check.
Author Response
We thank you for having reviewed our manuscript again and apologise for English and punctuation mistakes. The second author, who is a native English speaker, went through the paper again and made the necessary corrections. Most of the changes are highlighted in the text.